# Measuring Orthogonality in Representations of Generative Models

**Robin C. Geyer** *robin.geyer@inf.ethz.ch*
*Department of Computer Science, ETH Zurich*

**Alessandro Torcinovich** *alessandro.torcinovich@inf.ethz.ch*
*Faculty of Engineering, Free University of Bozen-Bolzano*
*Department of Computer Science, ETH Zurich*

**João B. Carvalho** *joao.carvalho@inf.ethz.ch*
*Department of Computer Science, ETH Zurich*

**Alexander Meyer** *alexander.meyer@dhzc-charite.de*
*German Heart Center of the Charité*

**Joachim M. Buhmann** *jbuhmann@inf.ethz.ch*
*Department of Computer Science, ETH Zurich*

**Reviewed on OpenReview:** *https://openreview.net/forum?id=TSUprKRga1*

## Abstract

In unsupervised representation learning, models aim to distill essential features from high-dimensional data into lower-dimensional learned representations, guided by inductive biases. Understanding the characteristics that make a good representation remains a topic of ongoing research. Disentanglement of independent generative processes has long been credited with producing high-quality representations. However, focusing solely on representations that adhere to the stringent requirements of most disentanglement metrics, may result in overlooking many high-quality representations, well suited for various downstream tasks. These metrics often demand that generative factors be encoded in distinct, single dimensions aligned with the canonical basis of the representation space.

Motivated by these observations, we propose two novel metrics: *Importance-Weighted Orthogonality (IWO)* and *Importance-Weighted Rank (IWR)*. These metrics evaluate the mutual orthogonality and rank of generative factor subspaces. Throughout extensive experiments on common downstream tasks, over several benchmark datasets and models, IWO and IWR consistently show stronger correlations with downstream task performance than traditional disentanglement metrics. Our findings suggest that representation quality is closer related to the orthogonality of independent generative processes rather than their disentanglement, offering a new direction for evaluating and improving unsupervised learning models.

## 1 Introduction

Humans are able to process rich data such as high-resolution images to distill and memorize key information about potentially complex concepts. Similarly, representation learning aims to devise a procedure, often unsupervised, to encode potentially high-dimensional data into a lower-dimensional learned embedding space, such that classifiers or other predictors can easily extract information from the learned representations (Bengio et al., 2013). Understanding how to generate these convenient representations, requires the definition of desirable properties that representation learning models should enforce. In the domain of generative models,

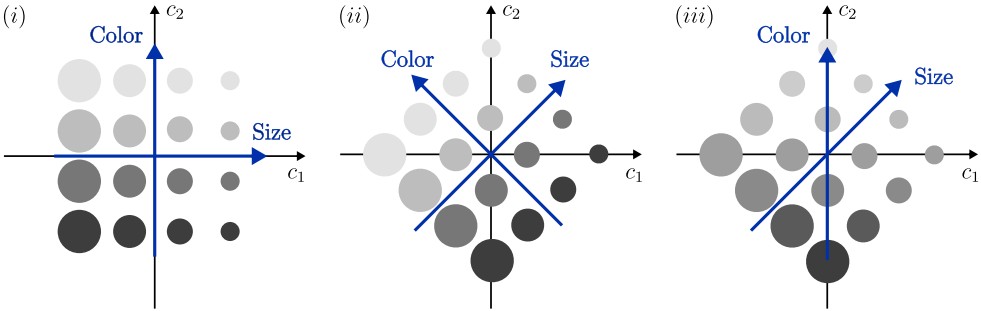

Figure 1: Three configurations of data (circles) encoded in a 2-d learned latent space. The data is characterized by size and (grayscale) color factors. The blue axes represent the direction of change in the factors. (i) The factors are aligned with the basis of the space, corresponding to perfect disentanglement and perfect orthogonality. (ii) The factors are not aligned but orthogonal, corresponding to complete entanglement, but still perfect orthogonality. (iii) The factors are not orthogonal and some circle configurations are not encoded, however, disentanglement is higher than in (ii), because of partial alignment with the basis. Despite its complete entanglement, we argue that latent space (ii) is just as well suited as (i) for common downstream tasks, while (iii) is not.

the ability to *disentangle* the explanatory factors underlying the data has long been credited to be such a desirable property.

In the disentangled representation learning framework, data $\boldsymbol{x}$ is often assumed to be generated by an underlying function $g$ driven by ground truth, generative factors $\{\boldsymbol{z}_j\}_{j=1}^{K}$ and other variability factors $\boldsymbol{t}$, that is $\boldsymbol{x} = g(\boldsymbol{z}, \boldsymbol{t})$. A model then learns a mapping $\boldsymbol{c} = r(\boldsymbol{x}) \in \mathbb{R}^L$ from the data to a latent representation space. A common characterization of disentanglement posits that the generative factors are represented by single distinct components of $\boldsymbol{c}$, implying that they manifest as orthogonal 1-d latent subspaces aligned with the canonical basis within the latent representation, up to a scaling factor and irrelevant latent dimensions (*i.e.*, when $L > K$).

Disentangled representations have played a pivotal role in improving the performance of tasks such as classification (Zhou et al., 2022), segmentation (Kalkhof et al., 2022), registration Han et al. (2020), image-to-image translation (Fei et al., 2021), artifact reduction (Tang et al., 2022), domain adaption (Li et al., 2021), controllable synthesis (Kelkar & Anastasio, 2022) and disease decomposition (Couronné et al., 2021). However, while disentanglement represents an intuitive and useful notion to characterize good representations, its general applicability is limited. Indeed, Locatello et al. (2019) prove, by using orthogonal transformations and probability (inverse) integral transforms, that unsupervised disentanglement learning is fundamentally impossible, without inductive biases on both models and datasets. In addition, the authors show that disentanglement exhibits weak correlation with the suitability of representations for common downstream tasks. We believe that this weak correlation can be attributed to the stringent nature of disentanglement measures, which penalize many high-quality representations otherwise suitable for downstream tasks.

To get an intuition of the problem, Figure 1 visualizes 2-d representation spaces encoding circles varying in size and color. While in space (i) and (ii) the dimensions encoding the generative factors are orthogonal to one another, in space (iii) they exhibit strong correlation. For the downstream task of regressing size and color, most models find the orthogonal spaces (i) and (ii) better suited than (iii), as they properly encode all size and color combinations, while space (iii) fails to encode the largest, lightest and the smallest, darkest circles. However, according to standard disentanglement metrics, space (ii) is completely entangled (worst case), while space (iii) exhibits a better disentanglement, due to the requirement that the generative factors *must align* with the canonical basis of the representation space. The disentanglement properties of these representations therefore stands in contrast to their downstream task utility.

While disentangled representations can be well-suited for many downstream tasks, entangled but orthogonal representations can be equally effective. In line with Wang & Isola (2020), we believe that correlation with

relevant downstream tasks is a necessary condition for any metric measuring representation utility. The absence of this correlation is a significant weakness of common disentanglement metrics.

We therefore propose two new metrics: *Importance-Weighted Orthogonality (IWO)* and *Importance-Weighted Rank (IWR)*. To compute these metrics, we devise a novel methodology, *Generative Component Analysis (GCA)*, that identifies the weighted vector subspaces where generative factors vary. IWO then measures the mutual weighted orthogonality between the subspaces found through GCA, while IWR assesses their weighted rank. Intuitively, IWO can be thought of as an expansion of cosine similarity to general vector subspaces.

We empirically assess the validity of the proposed metric in various synthetic experiments and by analysing the performance of three common downstream tasks across six benchmark datasets and six widely used models, showing that IWO and IWR consistently display stronger correlations with downstream task performance than popular disentanglement metrics such as *MIG* (Chen et al., 2018) or *DCI* (Eastwood & Williams, 2018). Our research suggests that the utility of a representation may be closer related to its orthogonality than its disentanglement.

## 2 Related Work

Alongside the task of disentanglement, gauging a model's performance in disentangling a representation has emerged as a non-trivial problem. Beyond visual inspection of the results, a variety of quantitative methodologies have been developed to tackle this issue. Higgins et al. (2017) propose to measure the accuracy of a classifier predicting the position of a fixed generative factor. Kim & Mnih (2018) further robustify the metric by proposing a majority voting scheme related to the least-variance factors in the representations. Chen et al. (2018) introduce the *Mutual Information Gap* estimating the normalized difference of the mutual information between the two highest factors of the representation vector. Eastwood & Williams (2018) propose the DCI metrics to evaluate the correlation between the representation and the generative factors. For each of them, one linear regressor is trained and the entropy over the rows (*Disentanglement*) and the columns (*Completeness*) is computed, along with the error (*Informativeness*) achieved by each regressor. The *Modularity* metric introduced by Ridgeway & Mozer (2018) computes the mutual information of each component of the representation to estimate its dependency with at most one factor of variation. *SAP* score (Kumar et al., 2018) estimates the difference, on average, of the two most predictive latent components for each factor.

The use of metrics such as the aforementioned ones contributed to shaping several definitions of disentanglement, each encoding a somewhat different aspect of disentangled representations, which led to a fragmentation of definitions (Locatello et al., 2019). Higgins et al. (2018) attempt instead to propose a unified view of the disentanglement problem, by defining a principled symmetry-based disentanglement framework, drawn from group representation theory. The authors established disentanglement in terms of a morphism from world states to decomposable latent representations, equivariant with respect to decomposable symmetries acting on the states/representations. For a representation to be disentangled, each symmetry group must act only on a corresponding (multidimensional) subspace of the representation. Following this conceptualization, Caselles-Dupré et al. (2019) demonstrate the learnability of such representations, provided the actions and the transitions between the states. Painter et al. (2020) extend the work by proposing a reinforcement learning pipeline to learn without the need for supervision. Noteworthy are the two proposed metrics: (i) an *independence score* that, similarly to our work, estimates the orthogonality between the generative factors in the fashion of a canonical correlation analysis; (ii) a *factor leakage score*, extended from the MIG metric to account for all the factors. Tonnaer et al. (2022) formalize the evaluation in the symmetry-based setting and proposed a principled metric that quantifies the disentanglement by estimating and applying the inverse group elements to retrieve an untransformed reference representation. A dispersion measure of such representations is then computed. Note that while most works focus on the linear manipulation of the latent subspace, the symmetry-based framework can also be used in non-linear cases. However, it requires modelling the symmetries and the group actions, a challenging task in scenarios with no clear underlying group structure (Tonnaer et al., 2022). We aim to develop a metric which does not require such a group structure and works in more general cases.

Recently, several works (Montero et al., 2021; Träuble et al., 2021; Dittadi et al., 2021) have proposed to go beyond the notion of disentanglement, advocating for the relaxation of the independence assumption among generative factors – perceived as too restrictive for real-world data problems – and modelling their correlations. Reddy et al. (2022) and Suter et al. (2019) formalize the concept of causal factor dependence, where the generative factors can be thought of as independent or subject to confounding factors. The latter work introduced the *Interventional Robustness Score* assessing the effects in the learned latent space when varying its related factors. Valenti & Bacciu (2022) define the notion of *weak disentangled* representation that leaves correlated generative factors entangled and maps such combinations in different regions of the latent space.

Additionally, Eastwood et al. (2023) relax the notion of disentanglement, by extending the DCI metric with an Explicitness (E) score related to the capacity required to regress the representation to its generative factors. Instead, our metric measures the mutual orthogonality between subspaces associated with generative factors, bypassing the non-linearities of a generative factor and its latent subspace. For a more comprehensive understanding refer to Appendix A.

## 3 Methodology

Our goal is to establish a metric that quantifies the total orthogonality of a representation. This involves estimating the orthogonality between the latent subspaces corresponding to each generative factor. However, this raises two challenges.

First, the identification of the latent subspaces is not straightforward since the generative factor can exhibit non-linear behaviour with respect to the learned representation. We propose *Generative Component Analysis (GCA)*, a procedure where, for each generative factor, multiple non-linear regressors are used to identify progressively smaller subspaces where most of the generative factor's information resides. These subspaces are then used to identify the *generative components*, that is, the set of orthonormal vectors spanning the latent subspace. In a similar fashion to procedures such as linear principal component regression, we weight each vector by a corresponding importance score to obtain an *importance-ordered orthogonal (i.o.o.)* basis, for each generative factor.

Second, prominent methods for measuring orthogonality between subspaces do not provide sufficiently discriminative results for our use case. The conventional definition of orthogonal complement is binary and too restrictive for nuanced applications. Conversely, methods like canonical correlation analysis, which determines the similarity between vector spaces, typically operate under optimal conditions and may assign high scores even to spaces sharing only a single dimension. These approaches, while useful, tend to be overly permissive for our specific objectives. We therefore propose *Importance-Weighted Orthogonality (IWO)* to characterize the orthogonality between generative factors. This is done by computing an average weighted projection of each generative factor's latent subspace onto all the others. To further discriminate between representations with similar orthogonality properties, we also compute *Importance-Weighted Rank (IWR)*, which estimates the spread of the importance weights, awarding representations that give more importance to a few dimensions.

### 3.1 Generative Component Analysis (GCA)

Consider a latent representation or *code*, $c \in \mathbb{R}^L$, encoding the generative factors $(z_1, \ldots, z_K) \in \mathbb{R}^K$, with $L \geq K$. The generative factors are inducded through $z_j = f_j^*(c)$, where $f_j^*$ is a potentially non-linear function. However, not all changes in $c$ imply a change in $z_j$. In particular, we define the *invariant latent subspace* of $z_j$ to be the largest linear subspace $\mathbb{I}_j \subseteq \mathbb{R}^L$, such that $f_j^*(c + v) = f_j^*(c)$, $\forall v \in \mathbb{I}_j$. Accordingly, the *variant latent subspace* (simply *latent subspace*) of $z_j$ is defined to be the orthogonal complement of $\mathbb{I}_j$ and will be denoted as $\mathbb{S}_j$, with dimensionality $R_j$. In the next paragraph, we describe how to find an importance-weighted basis for $\mathbb{S}_j$.

**Subspace learning**   Starting from a code $c \in \mathbb{R}^L$, we project it onto progressively smaller dimensional subspaces, removing the least important dimension for regressing $z_j$ at each step, until the subspace is

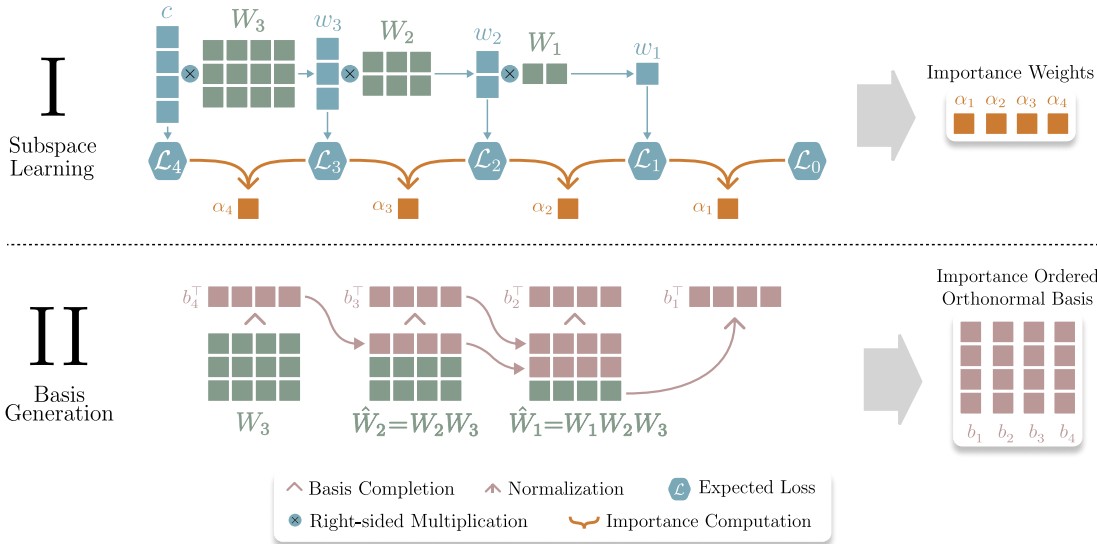

Figure 2: **Overview of GCA**. **I - Subspace Learning:** Through iterative multiplications with $\boldsymbol{W}_l \in \mathbb{R}^{l \times (l+1)}$, the input is projected to subspaces of decreasing dimensionality. The resulting outputs $\boldsymbol{w}_d$ are directed into NN heads, trained to minimize the expected loss terms $\mathcal{L}_l$. The importance $\alpha_l$ is gauged by the loss decrease between consecutive NNs heads. **II - Basis Generation:** The least important dimension, $\boldsymbol{b}_4$, corresponds to the null space of $\boldsymbol{W}_3$. For each subsequent dimension $\boldsymbol{b}_l$, the composed projection matrix $\hat{\boldsymbol{W}}_{l-1} = \boldsymbol{W}_{l-1} \cdot \dots \cdot \boldsymbol{W}_3$ is computed. $\boldsymbol{b}_l$ then corresponds to the dimension in the null space of $\hat{\boldsymbol{W}}_l$ which is orthogonal to all previously found basis vectors. Finally, $\boldsymbol{b}_1$ is retrieved by normalizing and transposing $\hat{\boldsymbol{W}}_1$.

1-dimensional. In particular, we design a Linear Neural Network (LNN) composed of a set of projective transformations $\boldsymbol{W}_{L-1}, \dots, \boldsymbol{W}_1$, with $\boldsymbol{W}_l \in \mathbb{R}^{l \times (l+1)}$, which reduce the dimensionality of $\boldsymbol{c}$ step-by-step. No non-linearities are applied, therefore each layer performs a projection onto a smaller linear subspace. The entire learning process is depicted in Figure 2.

To capture the most informative latent subspace of dimensionality $l$ at every layer of the LNN, we feed the intermediate projections $\boldsymbol{w}_l \in \mathbb{R}^l$ into non-linear neural networks $f_{jl}$. These networks are tasked with regressing $z_j$, thereby guiding the selection of which latent dimension to discard in each projection step.

When the training has ended, each regressor $f_{jl}$ can be associated with an expected loss of regressing the factor:

$$\mathcal{L}_l = \mathbb{E}_{\boldsymbol{c}} \left[ \ell(f_{jl}(\boldsymbol{w}_l(\boldsymbol{c})), z_j(\boldsymbol{c})) \right], \tag{1}$$

where $\ell$ is a specific loss term. In particular, note that $\mathcal{L}_{l-1} \geq \mathcal{L}_l$ because of the potential information loss due to dimensionality reduction. Let us now quantify the loss increment by each of the LNN projections as $\Delta \mathcal{L}_l = \mathcal{L}_{l-1} - \mathcal{L}_l$. We define $R_j$ as the smallest dimensionality at which the lowest achievable loss is reached, that is, $\Delta \mathcal{L}_l = 0$ for $l > R_j$. For $l = 1$, we compute $\Delta \mathcal{L}_1$ as the difference between $\mathcal{L}_1$ and a baseline loss $\mathcal{L}_0 = \mathbb{E}_{z_j} \left[ \ell(\mathbb{E}_{z_j} [z_j], z_j) \right]$.

**Basis generation** Using the trained projection matrices $\boldsymbol{W}_{L-1}, \dots, \boldsymbol{W}_1$, along with the layer-specific loss differences $\Delta \mathcal{L}_l$, we now describe how to construct an i.o.o. basis spanning $z_j$'s latent subspace $\mathbb{S}_j$. Formally, we want to devise a set of orthonormal vectors $B_j = \{\boldsymbol{b}_l^{(j)} \in \mathbb{R}^L \mid l = 1, \dots, R_j\}$, along with their respective importance weights $\{\alpha_l^{(j)} \in \mathbb{R}_{>0} \mid l = 1, \dots, R_j\}$. In the following, we drop the superscript $(j)$ to ease the notation.

Each projection matrix $\boldsymbol{W}_l$, $l = 1, \dots, L-1$ eliminates a dimension from the data representation. The adopted training methodology ensures that at each step the dimension discarded is the least important for regressing $z_j$ among the remaining ones. This is in turn based on which removed dimension results in the

minimum increase of $\Delta\mathcal{L}_{l+1}$. The removed dimension represents the 1-d null space of its corresponding projection matrix.

We begin by finding the basis vector $\boldsymbol{b}_L$, corresponding to the least important dimension, by identifying the null space of matrix $\boldsymbol{W}_{L-1} \in \mathbb{R}^{(L-1)\times L}$. For each subsequent dimension $\boldsymbol{b}_{l+1} \in \mathbb{R}^L$, $l = 1, \ldots, L-2$, we iteratively compute the composed projection matrix $\hat{\boldsymbol{W}}_l = \boldsymbol{W}_l \cdots \boldsymbol{W}_{L-1} \in \mathbb{R}^{l\times L}$. The null space $\ker(\hat{\boldsymbol{W}}_l)$ contains the target dimension $\boldsymbol{b}_{l+1}$ along with the previously found basis vectors $\boldsymbol{b}_{l+2}, \ldots, \boldsymbol{b}_L$. We retrieve $\boldsymbol{b}_{l+1}$ by finding the dimension in the null space of $\hat{\boldsymbol{W}}_l$ that is also orthogonal to the previously identified basis vectors, for example through QR decomposition. Finally, the remaining dimension $\boldsymbol{b}_1$, corresponding to the most significant basis vector, can be directly retrieved by computing $\hat{\boldsymbol{W}}_1 \in \mathbb{R}^{1\times L}$ and normalizing it. This process is depicted in Figure 2, and a pseudocode implementation can be found in Appendix E.

Using the loss difference defined on the basis of equation 1, we can quantify the importance weight $\alpha_l$ of each basis vector $\boldsymbol{b}_l$ by the relative loss increase associated to its layer in the LNN:

$$\alpha_l = \frac{\Delta\mathcal{L}_l}{\mathcal{L}_0 - \mathcal{L}_{R_j}} \qquad l = 1, \ldots, L. \tag{2}$$

Finally, we normalize each vector $\boldsymbol{b}_l$ thus obtaining an i.o.o. basis $B_j = \{\boldsymbol{b}_1^{(j)}, \ldots, \boldsymbol{b}_{R_j}^{(j)}\}$ for $\mathbb{S}_j$, with its corresponding importance weights $\alpha_1^{(j)}, \ldots, \alpha_{R_j}^{(j)}$.

GCA allocates an i.o.o. basis for each generative factor of a learned representation. However, when comparing representations, we also have to account for differences in $\mathcal{L}_{R_j}$, as this loss corresponds to the best possible regression of $z_j$ from the representation. In the DCI framework, this aspect is captured by the Informativeness metric. In order not to favour representations with low $R_j$ and high $\mathcal{L}_{R_j}$ over those with low $\mathcal{L}_{R_j}$ and higher $R_j$, we adjust the importance weights of any factor $z_j$ whose $\mathcal{L}_{R_j} > 0$. To do that, we first complete the factor's basis $B_j$ to span the whole latent space, then we distribute the loss $\mathcal{L}_{R_j}$ among the importance of the basis vectors equally:

$$\alpha_l = \frac{\Delta\mathcal{L}_l + \mathcal{L}_{R_j}/L}{\mathcal{L}_0} \qquad l = 1, \ldots, L. \tag{3}$$

### 3.2 Importance Weighted Orthogonality (IWO)

Orthogonality is commonly treated as a dichotomous attribute; that is, vectors are classified as either orthogonal or non-orthogonal, and similarly, a subspace is considered to either reside in the orthogonal complement of another or not. The concept of cosine similarity provides a continuous measure of the degree of orthogonality between two vectors. Analogously, we aim at a continuous measure that evaluates the degree of orthogonality between two subspaces. Consider two orthonormal bases $B_j = \{\boldsymbol{b}_1^{(j)}, \ldots, \boldsymbol{b}_{R_j}^{(j)}\}$ and $B_k = \{\boldsymbol{b}_1^{(k)}, \ldots, \boldsymbol{b}_{R_k}^{(k)}\}$ spanning two subspaces $\mathbb{S}_j$ and $\mathbb{S}_k$. Let $r_l^{(jk)}$ be the overall projection of $B_j$'s basis vector $\boldsymbol{b}_l^{(j)}$ onto all the basis vectors in $B_k$:

$$r_l^{(jk)} = \sum_{m=1}^{R_k} (\boldsymbol{b}_l^{(j)} \cdot \boldsymbol{b}_m^{(k)})^2 \qquad l = 1, \ldots, R_j, \tag{4}$$

where the square is applied to guarantee non-negativity. With these projections, a continuous interpretation of orthogonality between $\mathbb{S}_j$ and $\mathbb{S}_k$ can be expressed as

$$\mathrm{O}(\mathbb{S}_j, \mathbb{S}_k) = \frac{1}{\min(R_j, R_k)} \sum_{l=1}^{R_j} r_l^{(jk)}. \tag{5}$$

$\mathrm{O}(\mathbb{S}_j, \mathbb{S}_k) \in [0, 1]$ with its maximum reached if $\mathbb{S}_j$ is a subspace of $\mathbb{S}_k$ or vice-versa, and its minimum reached when $\mathbb{S}_k$ lies in the orthogonal complement of $\mathbb{S}_j$. This definition of orthogonality can be interpreted as the average squared cosine similarity between any vector pair from $\mathbb{S}_j$ and $\mathbb{S}_k$.

This formulation of orthogonality ignores the "importance" of dimensions within subspaces $\mathbb{S}_j$ and $\mathbb{S}_k$. To illustrate why this is problematic, consider two generative factors, $z_j$ and $z_k$, sharing the same subspace $\mathbb{S}$. If

most of their variation is concentrated in two dimensions, $\boldsymbol{b}_1^{(j)}$ and $\boldsymbol{b}_1^{(k)}$, while the rest of $\mathbb{S}$ encodes minimal variation, our metric should certainly distinguish between $\boldsymbol{b}_1^{(j)}$ and $\boldsymbol{b}_1^{(k)}$ being orthogonal vs. identical.

**Metrics**  As the name suggests, in addition to encapsulating the orthogonality between factor subspaces, IWO also takes into consideration the importance of the dimensions spanning them. Let us consider $K$ different generative factors $z_1, \ldots, z_K$. For each one of them, we are able to allocate an i.o.o. basis $B_k = \{\boldsymbol{b}_1^{(k)}, \ldots, \boldsymbol{b}_{R_k}^{(k)}\}$ with the importance weights $\{\alpha_1^{(k)}, \ldots, \alpha_{R_k}^{(k)}\}$. For any ground truth factor $z_j$, we define $\tilde{r}_l^{(jk)}$ as $\boldsymbol{b}_l^{(j)}$'s projection onto the latent subspace of another ground truth factor $z_k$, scaling the projection by the importance of the respective basis vectors:

$$\tilde{r}_l^{(jk)} = \sum_{m=1}^{R_k} \sqrt{\alpha_l^{(j)} \alpha_m^{(k)}} (\boldsymbol{b}_l^{(j)} \cdot \boldsymbol{b}_m^{(k)})^2 \qquad l = 1, \ldots, R_j. \tag{6}$$

Analogously to subspace orthogonality, IWO is then defined using the sum over all projections $\tilde{r}_l^{(jk)}$ of the dimensions spanning $z_j$'s subspace.

$$\text{IWO}(z_j, z_k) = 1 - \sum_{l=1}^{R_j} \tilde{r}_l^{(jk)}. \tag{7}$$

Note that, with respect to Equation 5, IWO does not require a normalization factor as the square root in Equation 6 guarantees that IWO $\in [0, 1]$. In addition, we subtract from 1 to simplify the comparison with standard disentanglement metrics. Therefore, the maximum of 1 is reached if $z_j$ lies in the orthogonal complement of $z_k$ and vice versa. The minimum of 0 is reached when $z_j$ and $z_k$, in addition to lying in the same subspace, also share the same importance along the same dimensions (*cf.* Appendix B for the proofs). Both Orthogonality and IWO can be efficiently calculated using matrix operations (*cf.* Appendix D).

Together with IWO, we define *Importance Weighted Rank (IWR)*. IWR measures how the importance of a generative factor's subspace is distributed among the dimensions spanning it:

$$\text{IWR}(z_j) = 1 - \mathcal{H}^{(j)}, \tag{8}$$

where $\mathcal{H}^{(j)} = -\sum_{l=1}^{R_j} \alpha_l^{(j)} \log_L(\alpha_l^{(j)})$ and we always assume $L > 1$. IWR thus measures how the importance is distributed among the $L$ dimensions of the representation space. Note that, $\text{IWR}(z_j) = 0$ if the importance is distributed equally along all $L$ dimensions spanning the representation space, while $\text{IWR}(z_j) = 1$ if the importance is concentrated in a single dimension only (*cf.* Appendix B for the proofs). We denote the mean over all generative factors of IWO and IWR as $\overline{\text{IWO}}$ and $\overline{\text{IWR}}$.

## 4  Experiments

To evaluate the effectiveness of our IWO and IWR implementations, we first test whether we can (i) recover the true latent subspaces using GCA and (ii) correctly assess their orthogonality and importance spread. For that purpose, we set up a synthetic data generation scheme, providing us with the ground truth IWO and IWR values. For comparison, we also test how other metrics capture different aspects of the representation. In particular, *Disentanglement*, *Completeness*, *Informativeness* and *Explicitness*, as measured by the DCI-ES framework using its official implementation.

We further evaluate IWO/IWR through the `disentanglement_lib` framework (Locatello et al., 2019) and measure how strong they correlate with downstream task performance for three different tasks deployed on the learned representations of six widely used variational autoencoder models trained on six benchmark disentanglement datasets and over a wide range of seeds and hyperparameters. More details are listed in the appendix and in our open-source code implementation[1].

Training of all neural networks in the LNN is performed in parallel. In principle, the gradient flow from each individual regressor $f_{jl}$ can be stopped after the corresponding $\boldsymbol{W}_l$, however, we attested a faster convergence

---

[1]https://github.com/cyrusgeyer/iwo

Table 1: Synthetic experimental results: comparison between (D) Disentanglement, (C) Completeness, (I) Informativeness, (E) Explicitness, $\overline{\text{IWO}}$ and $\overline{\text{IWR}}$ for Polynomial (Poly.) and Trigonometric (Trig.) encodings.

| Experiment ($K = 5$, always) | $L$ | $R$ | $f$ | ROP | D | C | I | E | $\overline{\text{IWO}}$ | $\overline{\text{IWR}}$ |
|---|---|---|---|---|---|---|---|---|---|---|
| (1) Permutation and additive Gaussian noise | 5 | 1 | Perm. | ✗ | 0.98 | 0.98 | 1.00 | 0.90 | 0.98 | 1.00 |
| | 5 | 1 | Noisy | ✗ | 0.98 | 0.98 | 1.00 | 0.90 | 0.98 | 1.00 |
| (2) Low rank + polynomial mapping | 10 | 2 | Poly. | ✗ | 0.99 | 0.69 | 1.00 | 0.75 | 0.98 | 0.69 |
| | 10 | 2 | Poly. | ✓ | 0.21 | 0.15 | 1.00 | 0.75 | 0.98 | 0.69 |
| (3) High rank + polynomial/trigonometric mapping | 10 | 5 | Poly. | ✗ | 0.41 | 0.30 | 1.00 | 0.74 | 0.61 | 0.31 |
| | 10 | 5 | Poly. | ✓ | 0.06 | 0.04 | 1.00 | 0.74 | 0.61 | 0.31 |
| | 10 | 5 | Trig. | ✗ | 0.41 | 0.30 | 0.99 | 0.73 | 0.62 | 0.30 |
| | 10 | 5 | Trig. | ✓ | 0.06 | 0.04 | 1.00 | 0.73 | 0.62 | 0.31 |
| (4) High rank + polynomial/trigonometric mapping | 20 | 4 | Poly. | ✗ | 0.98 | 0.53 | 0.99 | 0.76 | 0.97 | 0.54 |
| | 20 | 4 | Poly. | ✓ | 0.12 | 0.07 | 1.00 | 0.76 | 0.97 | 0.54 |
| | 20 | 4 | Trig. | ✗ | 0.96 | 0.52 | 0.99 | 0.73 | 0.98 | 0.53 |
| | 20 | 4 | Trig. | ✓ | 0.12 | 0.07 | 0.99 | 0.74 | 0.98 | 0.53 |
| | 20 | 8 | Poly. | ✗ | 0.56 | 0.30 | 0.99 | 0.74 | 0.76 | 0.31 |
| | 20 | 8 | Poly. | ✓ | 0.05 | 0.03 | 0.99 | 0.74 | 0.76 | 0.31 |
| | 20 | 8 | Trig. | ✗ | 0.54 | 0.29 | 0.98 | 0.71 | 0.76 | 0.31 |
| | 20 | 8 | Trig. | ✓ | 0.07 | 0.04 | 0.98 | 0.70 | 0.76 | 0.30 |
| (5) High dimensional latent space + poly./trig. mapping | 50 | 5 | Poly. | ✗ | 0.98 | 0.58 | 0.98 | 0.75 | 0.99 | 0.57 |
| | 50 | 5 | Poly. | ✓ | 0.11 | 0.05 | 0.98 | 0.75 | 0.99 | 0.56 |
| | 100 | 5 | Poly. | ✗ | 0.98 | 0.64 | 0.97 | 0.74 | 0.98 | 0.63 |
| | 100 | 5 | Poly. | ✓ | 0.09 | 0.04 | 0.98 | 0.73 | 0.98 | 0.63 |
| | 250 | 5 | Poly. | ✗ | 0.97 | 0.67 | 0.97 | 0.72 | 0.98 | 0.68 |
| | 250 | 5 | Poly. | ✓ | 0.09 | 0.04 | 0.97 | 0.72 | 0.98 | 0.68 |

when letting the gradient of each $f_{jl}$ flow back up to $c$. The aim of simultaneously exploring all nested subspaces is to facilitate the identification of the smallest subspaces by guidance through the larger ones. Additionally, we assume a reduction factor of 1, so that $W_l \in \mathbb{R}^{l \times (l+1)}$ for $l = 1, \ldots, L$, however, higher values can also be considered for larger representations. Further speed-up techniques are presented in Appendix I.

## 4.1 Synthetic Experiments

We introduce a synthetic data generating scheme, which generates vectors of i.i.d Gaussian distributed latent representations $c \in \mathbb{R}^L$. Then, on the basis of the latent representations, we synthesize $K$ generative factors $\{z_1, \ldots, z_K\}$. For simplicity, we choose $L$ as a multiple of $K$.

For simulating a disentangled latent space, we define each $z_j$ to be linearly dependent on a single, distinct element of $c$. To assess higher dimensional cases with non-linear relationships, we consider a non-linear commutative mapping $f : \mathbb{R}^{R_j} \to \mathbb{R}$. In particular, we experiment with a polynomial (Poly.) and a trigonometric (Trig.) $f$. Notice that the commutativity enforces that the distribution is spread evenly across all dimensions, such that we can easily assess the performance of $\overline{\text{IWR}}$. For simplicity, we always set $R_j = R$ for all $j = 1, \ldots, K$. Together, $L$ (latent space dimension), $K$ (number of factors) and $R$ (latent subspace dimension) determine how many dimensions each generative factor shares with the others. We consider the shared dimensions to be contiguous. For more details about the synthetic data generation scheme, refer to Figure 5 and the pseudocode in Appendix E.

To test representations that are not aligned with the canonical basis, we apply Random Orthogonal Projections (ROP) $R \in \mathbb{R}^{L \times L}$ to $c$. In line with commonly used datasets such as Cars3D or dSprites, we rescale and quantize $z_j$ to the range $[0, 1]$. All experiments are run four times with differing random seeds. The standard deviation was smaller than 0.02 for all reported values. The results are displayed in Table 1.

**(1) Permutation and additive Gaussian noise** We define two setups, both with $L = K = 5$ and $R = 1$. First, we let $z$ be a mere permutation of $c$, second we let $z$ be $c + \epsilon$, with $\epsilon \sim \mathcal{N}(0, 0.01)$ being Gaussian noise. As expected, under conditions of low-complexity and (quasi-)perfect disentanglement we obtain high scores for all the metrics.

**(2) Low rank + polynomial mapping** We choose $L = 10$, $K = 5$, $R = 2$. No dimensions are shared between the generative factors. The mapping $f$ is polynomial, adding more complexity. We also apply ROP in one of the experiments. We see that the $C$ score decreases due to dimension sharing. Additionally, ROP dramatically lower both $D$ and $C$, while $E$, $\overline{\text{IWO}}$ and $\overline{\text{IWR}}$ are resilient to it.

**(3) High rank + polynomial/trigonometric mapping** We set $L = 10$, $K = R = 5$ and vary ROP. We test with a polynomial and a trigonometric $f$. Each generative factor now shares, on average, two out of five dimensions with the others, determining a decrement in $D$, $C$, $\overline{\text{IWO}}$ and $\overline{\text{IWR}}$ scores. $E$ is slightly sensible to the change in the mapping used.

**(4) High rank + polynomial/trigonometric mapping** We set $L = 20$, $K = 5$, $R \in \{4, 8\}$, varying ROP and $f$. Increasing $L$ mitigates the mutual dependence between generative factors, with $\overline{\text{IWO}}$ and $\overline{\text{IWR}}$, partially recovering. In the presence of ROP, $D$ and $C$ keep low scores. $E$ stays almost constant assessing only the complexity of the function used.

**(5) High-dimensional latent space + polynomial/trigonometric mapping** We set $L \in \{50, 100, 250\}$, $K = R = 5$ and vary ROP, keeping $f$ polynomial. The large dimensionality of the representation mitigates the effect of dimension sharing. We observe that $\overline{\text{IWO}}$ and $\overline{\text{IWR}}$ assess good orthogonality reliably, while the other metrics perform similarly to the previous cases.

The experimental results reveal that both the $D$ and $C$ metrics from the DCI framework exhibit pronounced sensitivity to ROP, making them unsuitable for evaluating generative factor separability in the manner we propose. Although the $E$ metric from the DCI-ES framework demonstrates robustness against ROP, it still falls short of reliably capturing generative factor separability. This shortcoming arises because $E$ primarily measures the complexity of the mapping between generative factors rather than their actual separability. In contrast, our proposed metrics, $\overline{\text{IWO}}$ and $\overline{\text{IWR}}$, offer accurate assessments of orthogonality and the importance distribution across latent subspaces. They provide a more reliable evaluation of factor separability, demonstrating both resilience to orthogonal projections and immunity to mapping complexity.

## 4.2 Downstream Experiments

For the systematic evaluation of $\overline{\text{IWO}}$'s and $\overline{\text{IWR}}$'s correlation with downstream task performance, we use the `disentanglement_lib` framework (Locatello et al., 2019). We consider six benchmark datasets, namely **dSprites**, **Color dSprites** and **Scream dSprites** (Matthey et al., 2017), **Cars3D** (Reed et al., 2015), **smallNORB** (LeCun et al., 2004) and **Shapes3D** (Burgess & Kim, 2018). These datasets cover a wide range of complexities and variations, for more information on the individual datasets refer to Appendix F.2.

On each of these datasets, six different commonly used Variational Auto Encoder (VAE) models are trained: $\beta$-**VAE** Higgins et al. (2017), **Annealed VAE** (Burgess et al., 2018), $\beta$-**TCVAE** (Chen et al., 2018) **Factor-VAE** (Kim & Mnih, 2018), **DIP-VAE-I** and **DIP-VAE-II** (Kumar et al., 2018). These models represent a diverse set of approaches to disentanglement, each introducing unique mechanisms to encourage factorized representations. For each model, we consider six different regularization strengths (*cf.* Appendix F.1), each with ten different random seeds, resulting in a total of 2160 learned representations.

The representations are evaluated for their utility in regressing the generative factors. To this end, we consider three distinct downstream models trained on the learned representations: (i) random forest, (ii) logistic regression, and (iii) multi-layer perceptron. Correlations between downstream task performance and the commonly used metrics, DCI-D, DCI-C, and MIG, together with $\overline{\text{IWO}}$'s and $\overline{\text{IWR}}$, are calculated throughout the considered regularization strengths.

Our main objective is to investigate whether the orthogonality of a representation is indicative of the performance on a variety of downstream tasks and thus a good metric for the *utility or quality* of the representation, as elucidated by Wang & Isola (2020). The primary distinction between Orthogonality and Disentanglement, as measured by the considered metrics, is that the former does not require alignment with the canonical basis of the representation space. The three downstream models are chosen to distill this key

Table 2: (a) Average correlation between metrics and downstream task performance aggregated by model. Each model is trained on six different datasets. For each dataset six hyperparameters on 10 random seeds are trained for a total of 360 learned representations per model. (b) Average correlation between metrics and downstream task performance aggregated by dataset. For each dataset, six different VAE models are trained. For each model six hyperparameters on 10 random seeds are trained for a total of 360 learned representations per dataset.

(a) Per dataset correlation aggregated by model

| Model | DCI-D | DCI-C | MIG | IWO | IWR |
|---|---|---|---|---|---|
| (1) Random Forest | | | | | |
| AnnealedVAE | 0.74 | 0.64 | 0.66 | 0.74 | **0.82** |
| $\beta$-TCVAE | 0.66 | 0.63 | 0.64 | **0.78** | **0.66** |
| $\beta$-VAE | 0.65 | 0.65 | 0.64 | 0.26 | **0.83** |
| DIP-VAE-I | 0.56 | **0.59** | 0.52 | 0.43 | 0.40 |
| DIP-VAE-II | 0.47 | 0.32 | 0.44 | 0.09 | **0.66** |
| FactorVAE | 0.62 | 0.54 | 0.18 | **0.68** | 0.61 |
| (2) Logistic Regression | | | | | |
| AnnealedVAE | 0.43 | 0.20 | 0.26 | 0.77 | **0.85** |
| $\beta$-TCVAE | 0.13 | 0.06 | 0.13 | **0.45** | 0.09 |
| $\beta$-VAE | 0.18 | 0.19 | 0.16 | 0.44 | **0.51** |
| DIP-VAE-I | **0.73** | 0.63 | 0.59 | **0.73** | 0.63 |
| DIP-VAE-II | 0.07 | -0.53 | 0.01 | **0.88** | 0.16 |
| FactorVAE | 0.14 | -0.05 | 0.07 | **0.71** | 0.69 |
| (3) Multi-Layer Perceptron | | | | | |
| AnnealedVAE | 0.35 | 0.17 | 0.08 | 0.41 | **0.58** |
| $\beta$-TCVAE | -0.25 | -0.31 | -0.28 | -0.28 | -0.16 |
| $\beta$-VAE | -0.14 | -0.20 | -0.18 | **0.61** | 0.20 |
| DIP-VAE-I | 0.54 | 0.46 | 0.43 | **0.64** | 0.56 |
| DIP-VAE-II | 0.00 | 0.57 | 0.00 | **0.71** | 0.08 |
| FactorVAE | -0.04 | -0.19 | -0.39 | **0.51** | 0.44 |

(b) Per model correlation aggregated by dataset

| Model | DCI-D | DCI-C | MIG | IWO | IWR |
|---|---|---|---|---|---|
| (1) Random Forest | | | | | |
| Cars3d | -0.52 | -0.59 | -0.64 | **0.63** | 0.46 |
| Color dSprites | 0.73 | **0.79** | 0.69 | 0.33 | 0.77 |
| dSprites | 0.94 | **0.96** | 0.94 | 0.55 | 0.79 |
| Scream dSprites | 0.88 | **0.93** | 0.76 | 0.51 | 0.44 |
| Shapes3D | **0.88** | 0.76 | 0.66 | 0.13 | 0.63 |
| smallNORB | 0.78 | 0.54 | 0.67 | 0.82 | **0.88** |
| (2) Logistic Regression | | | | | |
| Cars3d | -0.39 | -0.57 | -0.70 | **0.75** | 0.69 |
| Color dSprites | **0.66** | 0.50 | 0.64 | 0.61 | 0.50 |
| dSprites | 0.38 | 0.35 | 0.30 | **0.58** | 0.36 |
| Scream dSprites | 0.79 | 0.55 | **0.86** | **0.86** | 0.67 |
| Shapes3D | -0.30 | -0.49 | -0.61 | **0.52** | -0.06 |
| smallNORB | 0.54 | 0.17 | 0.73 | 0.65 | **0.78** |
| (3) Multi-Layer Perceptron | | | | | |
| Cars3d | -0.12 | -0.36 | -0.39 | **0.56** | 0.63 |
| Color dSprites | -0.25 | -0.48 | -0.26 | -0.25 | -0.77 |
| dSprites | 0.18 | 0.17 | 0.13 | **0.35** | 0.08 |
| Scream dSprites | 0.18 | 0.01 | 0.23 | 0.59 | **0.78** |
| Shapes3D | -0.38 | -0.51 | -0.66 | **0.60** | 0.05 |
| smallNORB | 0.84 | 0.52 | 0.62 | 0.77 | **0.92** |

difference. While we expect alignment with the canonical basis to benefit downstream task (i) Random forest, we do not expect such a benefit for tasks (ii) logistic regression or (iii) multi-layer perceptron.

In Table 2a-2b, we present the results aggregated by model and dataset respectively. We refer the reader to the complete results in Table 3 in Appendix F for a detailed breakdown of all experiments and their outcomes. Figure 3 serves as an illustrative example, visualizing the comparison between the generative components as identified by GCA and the dimensions found through the DCI framework for one of the 2160 considered representation spaces.

**(1) Random forest**   High correlations are present for all models and datasets except for Cars3D, where the DCI and MIG metrics fail to correlate entirely. However, DCI-D correlates more reliably for the other datasets and models with the downstream task than IWO does. This can be attributed to the nature of random forests in defining axis-aligned discriminative rules. Therefore, the downstream task benefits from the alignment of the generative factors with the canonical basis. Indeed a random forest might have a harder time fitting latent space (ii) from Figure 1, than fitting latent space (iii), as in the latter, one generative factor aligns with the canonical basis. However, we also notice that DCI-C and IWR correlate even stronger than DCI-D and IWO do. This suggests that random forests benefit even more significantly from low-dimensional representations of generative factors than from their alignment with the canonical basis.

**(2) Logistic regression**   For logistic regression, where $\ell_2$-regularization is applied to the weights, there is no reason to assume that alignment between generative factors and canonical basis should lead to higher downstream task performance. Indeed, we can see that IWO and IWR correlate higher and more reliably than DCI or MIG do. This leads us to assume that measuring the orthogonality detached from canonical basis alignment provides a better metric for the evaluation of downstream logistic regression.

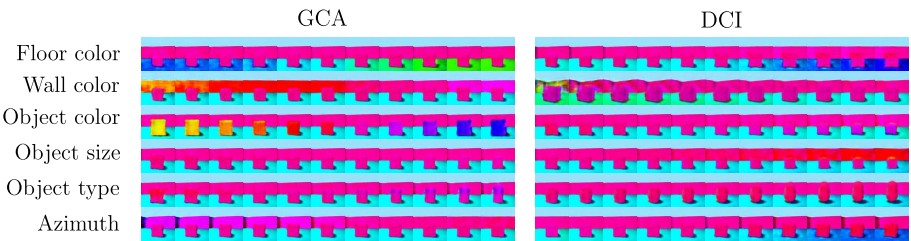

Figure 3: Samples from a $\beta$-VAE trained on Shapes3D dataset. In each row, the same latent code is modulated along a different dimension. The reconstruction of the resulting latent codes through the decoder network is depicted. **Left**: The dimensions of modulation correspond to the most important generative components of each generative factor as found by GCA. We attest that modulation along the generative components indeed predominantly varies the respective generative factor. **Right**: The dimension of modulation corresponds to the most important dimensions for each generative factor as found by DCI. However, modulation along the dimension supposedly encoding floor color also changes azimuth and vice versa. Modulation along the dimension supposedly encoding wall color also changes floor color. The same form of entanglement goes for most other dimensions identified through the DCI framework.

**(3) Multi-layer perceptron**  A similar behavior can be attested when fitting a multi-layer perceptron, initialized using Kaiming uniform initialization (He et al., 2015). There is, again, no reason to assume that alignment with the canonical basis would be beneficial. Indeed, we see that neither DCI-D, DCI-C nor MIG correlate reliably. IWO and IWR correlate reliably with this downstream task, except for the $\beta$-TCVAE model and the Color dSprites dataset.

## 5 Discussion and Conclusions

In our investigation, we pivoted from the conventional focus on the disentanglement of generative factors to a novel examination of their orthogonality. Our approach, geared towards accommodating non-linear behaviors within linear subspaces tied to generative factors, fills a gap in existing literature, as no prior method aptly addressed this perspective. The proposed Generative Component Analysis (GCA) efficiently identifies the generative factor subspaces and their importance. Leveraging GCA, we formulated Importance-Weighted Orthogonality (IWO) and Importance-Weighted Rank (IWR), two novel metrics offering unique insights into subspace orthogonality and importance ranking. Throughout experiments, our implementation emerged as a robust mechanism for assessing orthogonality, exhibiting resilience across varying latent shapes, non-linear encoding functions, and degrees of orthogonality.

Disentanglement has long been credited for fostering fairness, interpretability, and explainability in generated representations. However, as pointed out by Locatello et al. (2019), the utility of disentangled representations invariably hinges on at least partial access to generative factors. With such access, an orthogonal subspace could be rendered as useful as a disentangled one. Through orthonormal projection, any orthogonal representation discovered can be aligned with the canonical basis, achieving good disentanglement.

In conjunction with IWO's stronger downstream task correlation across datasets, models and tasks, this underscores our assertion that latent representations, which successfully decouple generative factors, are crucial for a wide range of downstream applications, regardless of their alignment with the canonical basis.

In conclusion, our work lays the groundwork for a fresh perspective on evaluating generative models. We hope GCA and IWO may help identify models crafting useful orthogonal subspaces, which might have been overlooked under the prevailing disentanglement paradigm. We hope that IWO extends its applicability across a broader spectrum of scenarios compared to traditional disentanglement measures.

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

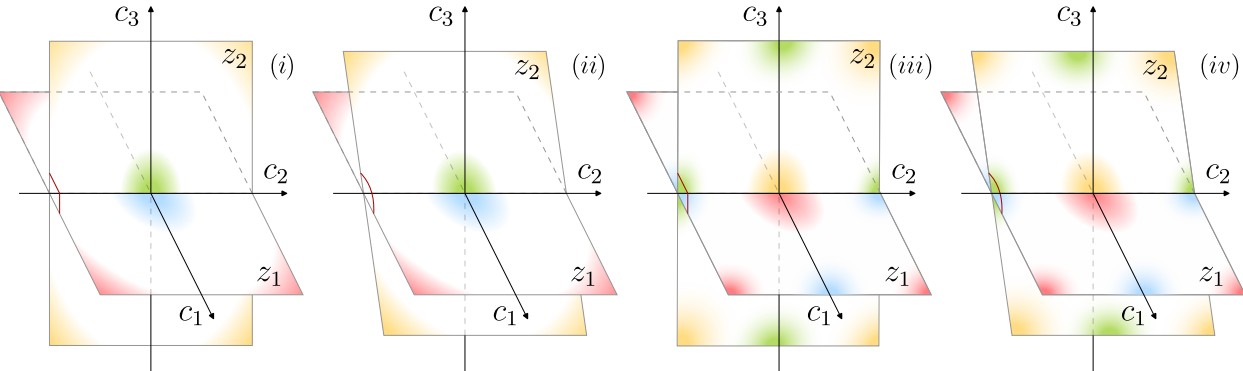

Figure 4: Four configurations of a 3-dimensional latent space. The planes represent the latent subspace where generative factors $z_1$, $z_2$ lie. The color mapping on each subspace represents the relationship between the generative factor and the latent components (*e.g.*, blue indicating large values for $z_1$, red indicating low ones). Cases (i) and (ii) are characterized by a good explicitness score as both subspaces encode $z_1$ and $z_2$ as simple quadratic functions, contrary to cases (iii) and (iv) where the relationship is trigonometric and more complex to recover. In contrast, cases (i) and (iii) are characterized by a better IWO score compared to (ii) and (iv). Indeed, in configurations (i) and (iii), there are dimensions within each generative factor's subspace that are orthogonal to one another. Consequently, any variation along any such dimension will leave the other factor unchanged.

## A  Explicitness vs IWO

The Explicitness (E) metric aims to evaluate the capacity needed for a representation to regress its generative factors. Formally,

$$E(z_j, \boldsymbol{c}; \mathcal{F}) = 1 - \frac{\text{AULCC}(z_j, \boldsymbol{c}; \mathcal{F})}{Z(z_j; \mathcal{F})}, \tag{9}$$

where $z_j$ and $\boldsymbol{c}$ represent a generative factor and the latent space respectively, and $\mathcal{F}$ a class of regressors (*e.g.*, multilayer perceptrons or random forests). AULCC is the Area Under the Loss Curve, computed by recording the minimum losses achievable by regressing $z_j$ from $\boldsymbol{c}$ with models in $\mathcal{F}$ of increasing capacity. The denominator, easily computable, acts as a normalizing constant so that $E \in [0, 1]$. As an example, $E = 1$ suggests that a linear regressor is sufficient to reach zero error, proving that the representation is efficient. To account for a bias toward large representations, the explicitness is paired with the Size (S), computed as the ratio between the number of generative factors and the size of the latent representation.

In Figure 4, we depict four situations of a 3-dimensional latent space where two generative factors $z_1$, $z_2$ lie in a separate 2-dimensional plane each, and the relationship between each generative factor and its corresponding latent subspace is non-linear, as hinted by the coloring. In particular, $z_j^{(n)} = f^{(n)}(\boldsymbol{c}') = f^{(n)}(\boldsymbol{P}_j^{(n)}\boldsymbol{c})$, for $j \in \{1, 2\}$ and $n \in \{i, ii, iii, iv\}$, with $\boldsymbol{P}_j^{(n)} \in \mathbb{R}^{2 \times 3}$ being a projection matrix. For cases (i) and (iii), the projections span orthogonal planes, contrarily to cases (ii) & (iv) where the planes have a different inclination. Case (i) & (ii) are characterized by a quadratic relationship, *i.e.*, $f^{(n)} = A(c_1')^2 + B(c_2')^2$ (with $A$, $B$ being parameters), while case (iii) and (iv) encode a trigonometric relationship, *i.e.*, $f^{(n)} = \cos(2\pi\lambda c_1') + \cos(2\pi\lambda c_2')$ (with $\lambda$ being a parameter).

Explicitness evaluates the capacity required by a model to regress the generative factors $z_1$, $z_2$, starting from the representation $\boldsymbol{c}$. Given that the effect of the linear transformation is present in all four situations, the differences are determined only by the non-linearity $f^{(n)}$. Therefore the metric is able to discriminate cases (i) & (ii) from (iii) & (iv). Instead, IWO quantifies the orthogonality of the planes, regardless of the non-linearities in the generative factors, so it discriminates the orthogonal cases (i) & (iii) from the non-orthogonal ones (ii) & (iv). Finally, note that the DCI-Disentanglement metric would penalize all four configurations as they are not disentangled.

# B    Theoretical Results on IWO/IWR

**Theorem B.1.** *Given a representation $\boldsymbol{c} \in \mathbb{R}^L$, $\overline{IWO} = 1$ if and only if the latent subspaces $\mathbb{S}_1, \ldots, \mathbb{S}_K$ of the generative factors $z_1, \ldots, z_K$ all lie in each other's orthogonal complement.*

*Proof.* Let $B_j$ and $B_k$ represent basis for the latent subspaces $\mathbb{S}_j$ and $\mathbb{S}_k$. If the subspaces lie in each other's orthogonal complement, this implies that $\boldsymbol{b}_l^{(j)} \cdot \boldsymbol{b}_m^{(k)} = 0$, $\forall l = 1, \ldots, R_j$, $\forall m = 1, \ldots, R_k$. The computation of IWO is therefore

$$\mathrm{IWO}(\boldsymbol{z}_j, \boldsymbol{z}_k) = 1 - \sum_{l=1}^{R_j} \sum_{m=1}^{R_k} \sqrt{\alpha_l^{(j)} \alpha_m^{(k)}} (\boldsymbol{b}_l^{(j)} \cdot \boldsymbol{b}_m^{(k)})^2 = 1 - \sum_{l=1}^{R_j} \sum_{m=1}^{R_k} \sqrt{\alpha_l^{(j)} \alpha_m^{(k)}} \cdot 0 = 1. \tag{10}$$

From proving that $\mathrm{IWO}(z_j, z_k) = 1$ for any $j \neq k$, it follows that $\overline{\mathrm{IWO}} = 1$.

Conversely, if at least one subspace does not lie in the orthogonal complement of at least one other subspace, this implies that $\boldsymbol{b}_l^{(j)} \cdot \boldsymbol{b}_m^{(k)} \neq 0$ for at least one combination of $l, m, j, k$. Because $\alpha_l^{(j)} > 0$ for $l \leq R_j$ for any $j$, this implies that $\mathrm{IWO}(\boldsymbol{z}_j, \boldsymbol{z}_k) < 1$. $\qquad \square$

**Theorem B.2.** *Given a representation $\boldsymbol{c} \in \mathbb{R}^L$, $\overline{IWO} = 0$ if and only if the latent subspaces $\mathbb{S}_1, \ldots, \mathbb{S}_K$ of the generative factors $z_1, \ldots, z_K$ all share the same importance along the same dimensions.*

*Proof.* Let $B_j$ and $B_k$ represent basis for the latent subspaces $\mathbb{S}_j$ and $\mathbb{S}_k$. From Section 3.1 we have that $\alpha_l^{(\zeta)} \geq \alpha_{l+1}^{(\zeta)}$ for any $l$ and any $\zeta \in \{j, k\}$. Let us first consider the special case $\alpha_l^{(\zeta)} > \alpha_{l+1}^{(\zeta)}$ for any $l$, i.e. the basis ordered from most important to least important basis vector. Assuming all subspaces share the same importance along the same dimensions, it must follow that $B_j = B_k$. Then $\boldsymbol{b}_l^{(j)} = \boldsymbol{b}_l^{(k)}$ and $\alpha_l^{(j)} = \alpha_l^{(k)}$, $\forall l$. Also, $\boldsymbol{b}_l^{(j)} \cdot \boldsymbol{b}_l^{(k)} = 1$ implies that $\boldsymbol{b}_l^{(j)} \cdot \boldsymbol{b}_m^{(k)} = 0$ for $l \neq m$. Then,

$$\begin{aligned}
\mathrm{IWO}(z_j, z_k) &= 1 - \sum_{l=1}^{R} \sum_{m=1}^{R} \sqrt{\alpha_l^{(j)} \alpha_m^{(k)}} (\boldsymbol{b}_l^{(j)} \cdot \boldsymbol{b}_m^{(k)})^2 \\
&= 1 - \sum_{l=1}^{R} \sqrt{\alpha_l^{(j)} \alpha_l^{(k)}} \\
&= 1 - \sum_{l=1}^{R} \alpha_l^{(j)} \\
&= 1 - 1 = 0.
\end{aligned} \tag{11}$$

Let us now consider the special case where $\mathbb{S}_j = \mathbb{S}_k = \mathbb{R}^R$ and $\alpha_l^{(j)} = \alpha_m^{(k)} = 1/R$, $\forall l, m$. Note that, since all importance weights are equal, there is no order in the basis. Let $B_j$ and $B_k$ be two orthogonal bases spanning $\mathbb{S}_j$ and $\mathbb{S}_k$ respectively. Because $\mathbb{S}_j = \mathbb{S}_k$, each basis vector $\boldsymbol{b}_l^{(j)}$ in $B_j$ can be expressed as a linear combination of the basis vectors in $B_k$. We can then use Parseval's identity (Johnson & Riess, 1982) for the

finite case: $\|\boldsymbol{b}_l^{(j)}\|^2 = \sum_{m=1}^R (\boldsymbol{b}_l^{(j)} \cdot \boldsymbol{b}_m^{(k)})^2$.

$$\begin{aligned}
\text{IWO}(z_j, z_k) &= 1 - \sum_{l=1}^R \sum_{m=1}^R \sqrt{\alpha_l^{(j)} \alpha_m^{(k)}} (\boldsymbol{b}_l^{(j)} \cdot \boldsymbol{b}_m^{(k)})^2 \\
&= 1 - \sum_{l=1}^R \sum_{m=1}^R \sqrt{\frac{1}{R} \cdot \frac{1}{R}} (\boldsymbol{b}_l^{(j)} \cdot \boldsymbol{b}_m^{(k)})^2 \\
&= 1 - \frac{1}{R} \sum_{l=1}^R \sum_{m=1}^R (\boldsymbol{b}_l^{(j)} \cdot \boldsymbol{b}_m^{(k)})^2 \\
&= 1 - \frac{1}{R} \sum_{l=1}^R \|\boldsymbol{b}_l^{(j)}\|^2 = 1 - \frac{R}{R} = 0.
\end{aligned} \tag{12}$$

The general case for $\alpha_l^{(j)} \geq \alpha_{l+1}^{(j)}$ then follows from the two special cases described above. From proving that $\text{IWO}(z_j, z_k) = 0$ for any $j \neq k$, it follows that $\overline{\text{IWO}} = 0$.

Conversely, if two generative factors do not share the same importance along the same dimensions, this can either mean that their subspaces are different or it means their subspaces are the same, but the importance is spread differently, i.e. $\alpha_l^{(j)} \neq \alpha_l^{(k)}$ for at least two different $l$. For both cases it follows that $\text{IWO}(z_j, z_k)$ cannot be 0. $\qquad \square$

**Theorem B.3.** *Given a representation $\boldsymbol{c} \in \mathbb{R}^L$ with $L > 1$, then $\overline{IWR} = 1$ if and only if the latent subspaces $\mathbb{S}_1, \ldots, \mathbb{S}_K$ of the generative factors $z_1, \ldots, z_K$ are all uni-dimensional.*

*Proof.* Let $B_j = \{\boldsymbol{b}_1^{(j)}\}$, then $\alpha_1^{(j)} = 1$ and

$$\text{IWR}(z_j) = 1 + \sum_{l=1}^1 \alpha_l^{(j)} \log_L(\alpha_l^{(j)}) = 1 + 1 \log_L(1) = 1 + 0 = 1, \tag{13}$$

From proving that $\text{IWR}(z_j) = 1$ for any $j$, it follows that $\overline{\text{IWR}} = 1$.

Conversely, if $\alpha_l^{(j)} > 0$ for $l > 1$, under the constraint that $\alpha_l^{(j)} \geq \alpha_{l+1}^{(j)}$ and $\sum_l^{R_j} \alpha_l^{(j)} = 1$, then $\text{IWR}(z_j)$ cannot be 1. $\qquad \square$

**Theorem B.4.** *Given a representation $\boldsymbol{c} \in \mathbb{R}^L$ with $L > 1$, then $\overline{IWR} = 0$ if and only if for each generative factor $z_1, \ldots, z_K$ the importance is spread equally among all dimensions of the representation space.*

*Proof.* Let $B_j = \{\boldsymbol{b}_1, \ldots, \boldsymbol{b}_L\}$ and $\alpha_l^{(j)} = 1/L$, $l = 1, \ldots, L$, then:

$$\text{IWR}(z_j) = 1 + \sum_{l=1}^L \alpha_l^{(j)} \log_L \alpha_l^{(j)} = 1 + \sum_{l=1}^L \frac{1}{L} \log_L \left(\frac{1}{L}\right) = 1 + \sum_{l=1}^L \frac{1}{L} \cdot (-1) = 1 - 1 = 0. \tag{14}$$

From proving that $\text{IWR}(z_j) = 0$ for any $j$, it follows that $\overline{\text{IWR}} = 0$.

Conversely if any $\alpha_l^{(j)} \neq 1/L$, under the constraint that $\sum_l^L \alpha_l^{(j)} = 1$, then $\text{IWR}(z_j)$ cannot be 0. $\qquad \square$

## C  Complexity and Differentiability

For GCA, the subspace learning step complexity is dependent on the backpropagation algorithm used, while the basis generation step complexity is dependent on the reduced QR decomposition algorithm used. The worst-case complexity of $\overline{\text{IWO}}$ is $O(KL^3)$, for $\overline{\text{IWR}}$ it is $O(KL)$. GCA and the metrics are composed of differentiable operations and are therefore differentiable. While the metrics are optimizable, GCA requires supervision from the generative factors, information that may not always be available.

# D    Efficient IWO Calculation

For the efficient calculation of Orthogonality, consider two matrices $\boldsymbol{B}_j \in \mathbb{R}^{R_j \times L}$, $\boldsymbol{B}_k \in \mathbb{R}^{R_k \times L}$ whose rows compose the i.o.o. basis vectors spanning $z_j$'s and $z_k$'s latent subspaces respectively. We define the orthogonality between the two latent subspaces in terms of these matrices as:

$$O(\mathbb{S}_j, \mathbb{S}_k) = \frac{\text{Tr}(\boldsymbol{B}_j \boldsymbol{B}_k^\top \boldsymbol{B}_k \boldsymbol{B}_j^\top)}{\min(R_j, R_k)}. \tag{15}$$

Note that the trace $\text{Tr}(\boldsymbol{B}_j \boldsymbol{B}_k^\top \boldsymbol{B}_k \boldsymbol{B}_j^\top)$ equals the sum of the squared values in $\boldsymbol{B}_j \boldsymbol{B}_k^\top$, i.e., $\sum_{l,m}(\boldsymbol{B}_j \boldsymbol{B}_k^\top)_{ml}^2 = \sum_{l,m}(\boldsymbol{b}_{jl} \cdot \boldsymbol{b}_{km})^2$, where $\boldsymbol{b}_{jl}$ and $\boldsymbol{b}_{km}$ are the $l$-th and $m$-th rows of $\boldsymbol{B}_j$ and $\boldsymbol{B}_k$ respectively. The maximum of the trace is therefore $\min(R_j, R_k)$, reached if $\mathbb{S}_j$ is a subspace of $\mathbb{S}_k$ or vice-versa. This definition of orthogonality can be interpreted as the average absolute cosine similarity between any vector pair from $\mathbb{S}_j$ and $\mathbb{S}_k$.

To efficiently calculate the importance-weighted projection of $z_j$'s subspace onto $z_k$'s subspace, we first scale the corresponding bases vectors in $\boldsymbol{B}_j, \boldsymbol{B}_k$ with their respective importance before projecting them onto one another. IWO is the sum of all individual projections. Using $\boldsymbol{U}_j = \boldsymbol{D}_j \boldsymbol{B}_j$, where $\boldsymbol{D}_j \in \mathbb{R}^{R_j \times R_j}$ is diagonal with the $l$-th diagonal entry corresponding to the square root of importance, $\sqrt{\alpha_l(z_j)}$, we can efficiently calculate IWO as:

$$\text{IWO}(z_j, z_k) = 1 - \text{Tr}(\boldsymbol{U}_j \boldsymbol{U}_k^\top \boldsymbol{U}_k \boldsymbol{U}_j^\top) \tag{16}$$

# E    Pseudocode

## E.1    Basis Generation

**Full QR decomposition**    Given a rectangular matrix $\boldsymbol{A} \in \mathbb{R}^{n \times (n-1)}$, its full QR decomposition is

$$\boldsymbol{A} = \begin{bmatrix} \boldsymbol{Q} & \boldsymbol{v} \end{bmatrix} \begin{bmatrix} \boldsymbol{R} \\ \boldsymbol{0} \end{bmatrix}, \tag{17}$$

with $\boldsymbol{R} \in \mathbb{R}^{(n-1) \times (n-1)}$ being upper-triangular, and the columns of $\boldsymbol{Q} \in \mathbb{R}^{n \times (n-1)}$ and $\boldsymbol{v} \in \mathbb{R}^{n \times 1}$ being orthonormal. In particular, note that $\boldsymbol{A}^\top \boldsymbol{v} = \boldsymbol{0}$, that is $\boldsymbol{v} \in \ker(\boldsymbol{A}^\top)$. We denote $\boldsymbol{v} = \text{QR}_{\text{ker}}(\boldsymbol{A})$.

We report the pseudocode for generating an orthonormal basis as described in Section 3

**Require:** Projection matrices $\{\boldsymbol{W}_l\}_{l=1}^{L-1}$, $\boldsymbol{W}_l \in \mathbb{R}^{l \times (l+1)}$
**Ensure:** Orthonormal basis matrix $\boldsymbol{B} \in \mathbb{R}^{L \times L}$

1:  $\hat{\boldsymbol{W}} \leftarrow \boldsymbol{W}_{L-1}^\top$                                      ▷ $\hat{\boldsymbol{W}} \in \mathbb{R}^{L \times (L-1)}$
2:  $\boldsymbol{b}_L \leftarrow \text{QR}_{\text{ker}}(\hat{\boldsymbol{W}})$                          ▷ Get least important basis vector $\boldsymbol{b}_L$.
3:  $\boldsymbol{B} \leftarrow \boldsymbol{b}_L$
4:  **for** $l \leftarrow L-2$ **to** 2 **do**
5:      $\hat{\boldsymbol{W}} \leftarrow \hat{\boldsymbol{W}} \cdot \boldsymbol{W}_l^\top$                      ▷ $\hat{\boldsymbol{W}} \in \mathbb{R}^{L \times l}$
6:      $\tilde{\boldsymbol{W}} \leftarrow [\hat{\boldsymbol{W}}^\top \ \boldsymbol{B}]$                ▷ Concatenate $\hat{\boldsymbol{W}}^\top$ and $\boldsymbol{B}$. $\tilde{\boldsymbol{W}} \in \mathbb{R}^{L \times (L-1)}$
7:      $\boldsymbol{b}_l \leftarrow \text{QR}_{\text{ker}}(\tilde{\boldsymbol{W}})$                        ▷ Get basis vector $\boldsymbol{b}_l$
8:      $\boldsymbol{B} \leftarrow [\boldsymbol{b}_l \ \boldsymbol{B}]$                                  ▷ Concatenate $\boldsymbol{b}_l$ to $\boldsymbol{B}$
9:  **end for**
10: $\hat{\boldsymbol{W}} \leftarrow (\boldsymbol{W}_1 \cdot \hat{\boldsymbol{W}})^\top$                    ▷ $\hat{\boldsymbol{W}} \in \mathbb{R}^{L \times 1}$
11: $\boldsymbol{b}_1 \leftarrow \frac{\hat{\boldsymbol{W}}}{\|\hat{\boldsymbol{W}}\|_2}$                    ▷ Get most important basis vector $\boldsymbol{b}_1$
12: $\boldsymbol{B} \leftarrow [\boldsymbol{b}_1 \ \boldsymbol{B}]$                                  ▷ Concatenate $\boldsymbol{b}_1$ to $\boldsymbol{B}$
13: **return** $\boldsymbol{B}$

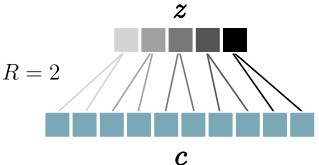 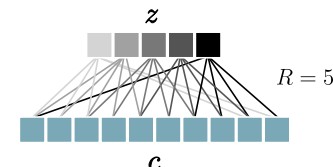

Figure 5: Two synthetic experimental settings with $L = 10$, $K = 5$ and differing ranks $R$. **Left**: $R = 2$, each $z_j$ is a function of two successive elements of $\boldsymbol{c}$: $z_1 = f(c_1, c_2), \ldots, z_5 = f(c_9, c_{10})$. **Right**: $R = 5$, each $z_j$ is a function of five successive elements of $\boldsymbol{c}$: $z_1 = f(c_9, c_{10}, c_1, c_2, c_3) , \ldots, z_5 = f(c_7, c_8, c_9, c_{10}, c_1)$.

### E.2 Synthetic Data Generation

We report the pseudocode for generating the synthetic representations discussed in Section 4.1. For simplicity, we have assumed $L$ as a multiple of $K$. Figure 5 describes two examples of the used schema.

**Require:** Representation size $L$, # of generative factors $K$, # of shared dimensions $R$, mapping $f$
**Ensure:** A representation $\boldsymbol{c} \in \mathbb{R}^L$ and its related generative factors $\boldsymbol{z} \in \mathbb{R}^K$

1: $\boldsymbol{c} \leftarrow \mathcal{N}(\boldsymbol{0}_L, \boldsymbol{I}_L)$           ▷ Get $L$ i.i.d. Gaussian distributed samples
2: $\boldsymbol{c}^{(\mathrm{rep})} \leftarrow \mathrm{cat}(\boldsymbol{c}, 3)$           ▷ Concatenate $\boldsymbol{c}$ three times, $\boldsymbol{c}^{(rep)} \in \mathbb{R}^{3L}$
3: $l_d \leftarrow \lfloor \frac{R}{2} \rfloor$
4: $l_u \leftarrow \lceil \frac{R}{2} \rceil$
5: $\boldsymbol{z} \leftarrow [\ ]$
6: **for** $l \leftarrow L$ to $2L$ **by** $\frac{L}{K}$ **do**
7:      $\boldsymbol{x} \leftarrow \boldsymbol{c}^{(\mathrm{rep})}_{l - l_d : l + l_u}$           ▷ Get values of $\boldsymbol{c}^{(\mathrm{rep})}$ from $l - l_d$ to $l + l_u$
8:      $\boldsymbol{z} \leftarrow [\boldsymbol{z}\ f(\boldsymbol{x})]$           ▷ Concatenate the generative factor $f(\boldsymbol{x})$ to $\boldsymbol{z}$
9: **end for**
10: **return** $\boldsymbol{c}, \boldsymbol{z}$

## F   Experimental Details

In this section, we provide a detailed description of the correlation analysis of our orthogonality metric with downstream tasks on six different datasets and six different models listed in Table 3. For each model, learned representations for six different regularization strengths are considered (ten different random seeds for each reg. strength). All these representations are directly retrieved from or trained with the code of `disentanglement_lib`[2]. For the ES metric, we utilized the official codebase provided by the authors of DCI-ES [3]. Each dataset we investigate has independent generative factors associated with it.

### F.1   Hyperparameters

The hyperparameters considered for each model are the following:

- **$\beta$-VAE** (Higgins et al., 2017): with $\beta \in \{1, 2, 4, 6, 8, 16\}$

- **Annealed VAE** (Burgess et al., 2018): with $c_{max} \in \{5, 10, 25, 50, 75, 100\}$

- **$\beta$-TCVAE** (Chen et al., 2018): with $\beta \in \{1, 2, 4, 6, 8, 10\}$

- **Factor-VAE** (Kim & Mnih, 2018): with $\gamma \in \{10, 20, 30, 40, 50, 100\}$

- **DIP-VAE-I** (Kumar et al., 2018): with $\lambda_{od} \in \{1, 2, 5, 10, 20, 50\}$

- **DIP-VAE-II** (Kumar et al., 2018): with $\lambda_{od} \in \{1, 2, 5, 10, 20, 50\}$

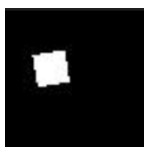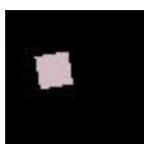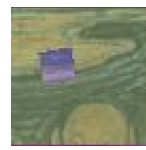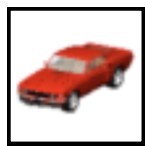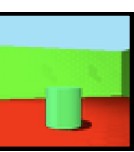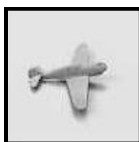

Figure 6: Samples from the Datasets used in our studies. From left to right: DSprites, Color DSprites, Scream DSprites, Cars3D, Shapes3D, smallNORB

## F.2 Datasets

**dSprites Dataset**  The dSprites dataset is a collection of 2D shape images procedurally generated from six independent latent factors. These factors are color (white), shape (square, ellipse, heart), scale, rotation, and x and y positions of a sprite. Each possible combination of these latents is present exactly once, resulting in a total of 737280 unique images.

**Color dSprites Dataset**  This dataset retains the fundamental characteristics of the original dSprites dataset, with the distinct variation that each sprite, in the observation sampling process, is rendered in a color determined by random selection.

**Scream dSprites Dataset**  The dataset mirrors the original dSprites dataset but introduces a unique modification in the observation sampling process: A random segment from the Scream image is selected as the background, and the sprite is integrated into this image by inverting the color of the chosen segment specifically at the sprite's pixel locations.

**Cars3D Dataset**  The Cars3D dataset is generated from 3D computer-aided design (CAD) models of cars. It consists of color renderings of 183 car models from 24 rotation angles, each offset by 15 degrees, and from 4 different camera elevations. The images are rendered at a resolution of $64 \times 64$.

**smallNORB Dataset**  The smallNORB dataset is designed for 3D object recognition from shape, featuring images of 50 toys categorized into five types: four-legged animals, human figures, airplanes, trucks, and cars. The images were captured under six lighting conditions, nine elevations, and 18 different angles. The dataset is split into a training set with five instances of each category and a test set with the remaining five instances.

**Shapes3D Dataset**  The Shapes3D dataset, initially presented in Kim & Mnih (2018) is a specialized collection designed for the study of factorized variations in 3D object representation. This dataset is characterized by its systematic variation across six ground-truth factors, making it particularly suitable for experiments in disentanglement and representation learning. These factors include floor color with 10 variations, wall color also with 10 variations, object color featuring 10 distinct options, object size represented in 8 different scales, object type with 4 unique categories, and azimuth with 15 varied positions. Such a diverse range of factors allows for comprehensive analysis and experimentation in 3D object recognition and disentanglement tasks. The dataset is thoughtfully split into a training set and a test set, each containing a balanced mix of these variations to facilitate robust model training and evaluation.

## F.3 IWO on Limited Data

To assess the impact of smaller sample sizes on IWO and IWR's correlation with downstream task performance, we repeat the experiments detailed in Section 4.2 for the smallNORB dataset, but with only 50% and 10% of the data. The results are illustrated in Table 4. We observe that IWO and IWR are resilient to changes in dataset size.

---

[2]https://github.com/google-research/disentanglement_lib/tree/master
[3]https://github.com/andreinicolicioiu/DCI-ES

Table 3: Individual correlations between metrics and downstream task performance. Each model is trained on six different datasets. For each dataset, six different hyperparameters on 10 random seeds are trained for a total of 2160 learned representations. Correlations between downstream task performance and the three commonly used metrics DCI-D, DCI-C and MIG together with IWO and IWR are calculated. Correlations are calculated between averaged task performance and averaged metrics, where the average is taken over the seeds and grouped by hyperparameter. As such, we assess the capability of the metrics to point us to good models and hyperparameters.

| | | Random Forest | | | | | Logistic Regression | | | | | Multi-Layer Perceptron | | | | |
|---|---|---|---|---|---|---|---|---|---|---|---|---|---|---|---|---|
| | | D | C | MIG | IWO | IWR | D | C | MIG | IWO | IWR | D | C | MIG | IWO | IWR |
| Cars3D | AnnealedVAE | 0.72 | -0.18 | -0.19 | 0.74 | 0.69 | -0.40 | -0.85 | -0.82 | 0.84 | 0.89 | 0.25 | -0.36 | -0.33 | 0.98 | 0.97 |
| | β-TCVAE | -0.92 | -0.91 | -0.91 | 0.80 | 0.11 | -0.94 | -0.97 | -0.90 | 0.75 | -0.15 | 0.06 | -0.04 | 0.16 | -0.48 | -0.90 |
| | β-VAE | -0.92 | -0.94 | -0.93 | 0.67 | 0.80 | -0.93 | -0.95 | -0.92 | 0.60 | 0.91 | -0.66 | -0.67 | -0.67 | 0.22 | 0.82 |
| | DIP-VAE-I | -0.84 | -0.77 | -0.61 | -0.09 | -0.18 | 0.61 | 0.39 | -0.10 | 0.68 | 0.73 | -0.07 | -0.32 | -0.50 | 0.98 | 0.99 |
| | DIP-VAE-II | -0.89 | -0.96 | -0.76 | 0.90 | 0.59 | -0.49 | -0.72 | -0.74 | 0.74 | 0.83 | -0.49 | -0.70 | -0.49 | 0.67 | 0.94 |
| | FactorVAE | -0.26 | -0.17 | -0.81 | 0.78 | 0.74 | -0.17 | -0.31 | -0.71 | 0.91 | 0.91 | 0.22 | -0.08 | -0.49 | 0.97 | 0.98 |
| Color dSp. | AnnealedVAE | 1.00 | 1.00 | 0.98 | 0.97 | 0.97 | 0.86 | 0.85 | 0.90 | 0.91 | 0.94 | -0.80 | -0.80 | -0.82 | -0.86 | -0.85 |
| | β-TCVAE | 0.99 | 0.99 | 1.00 | 0.78 | 0.97 | 0.44 | 0.54 | 0.45 | 0.70 | 0.55 | -0.81 | -0.89 | -0.85 | -0.68 | -0.86 |
| | β-VAE | 0.98 | 0.99 | 1.00 | -0.80 | 0.76 | 0.69 | 0.85 | 0.80 | -0.45 | 0.93 | -0.67 | -0.85 | -0.78 | 0.32 | -0.95 |
| | DIP-VAE-I | 0.46 | 0.39 | 0.61 | 0.96 | 0.94 | 0.73 | 0.68 | 0.70 | 0.78 | 0.63 | 0.89 | 0.91 | 0.75 | 0.04 | -0.18 |
| | DIP-VAE-II | 0.07 | 0.50 | -0.07 | -0.73 | 0.26 | 0.37 | -0.78 | 0.39 | 0.90 | -0.81 | 0.71 | -0.45 | 0.73 | 0.50 | -0.92 |
| | FactorVAE | 0.91 | 0.85 | 0.63 | 0.81 | 0.75 | 0.86 | 0.86 | 0.62 | 0.82 | 0.72 | -0.79 | -0.80 | -0.60 | -0.80 | -0.87 |
| dSprites | AnnealedVAE | 0.99 | 0.99 | 0.97 | 0.89 | 0.89 | 0.84 | 0.82 | 0.85 | 0.87 | 0.87 | 0.47 | 0.45 | 0.54 | 0.73 | 0.72 |
| | β-TCVAE | 1.00 | 0.99 | 0.99 | 0.92 | 0.97 | 0.37 | 0.38 | 0.25 | 0.66 | 0.54 | -0.14 | -0.18 | -0.25 | 0.07 | 0.03 |
| | β-VAE | 0.98 | 0.99 | 0.97 | 0.75 | 0.96 | 0.03 | 0.07 | 0.01 | -0.18 | 0.02 | -0.04 | -0.06 | 0.02 | 0.40 | 0.08 |
| | DIP-VAE-I | 0.96 | 0.96 | 0.94 | 0.28 | 0.11 | 0.80 | 0.79 | 0.77 | 0.44 | 0.35 | 0.91 | 0.91 | 0.90 | -0.10 | -0.22 |
| | DIP-VAE-II | 0.77 | 0.88 | 0.80 | -0.33 | 0.93 | -0.20 | -0.46 | -0.43 | 0.69 | -0.59 | -0.92 | -1.00 | -0.92 | 0.14 | -0.98 |
| | FactorVAE | 0.96 | 0.94 | 0.96 | 0.81 | 0.86 | 0.43 | 0.48 | 0.35 | 0.99 | 0.98 | 0.79 | 0.93 | 0.49 | 0.86 | 0.84 |
| Scream dSp. | AnnealedVAE | 0.84 | 0.79 | 0.85 | 0.86 | 0.87 | 0.92 | 0.85 | 0.95 | 0.92 | 0.91 | 0.81 | 0.86 | 0.71 | 0.84 | 0.84 |
| | β-TCVAE | 0.91 | 0.98 | 0.90 | 0.32 | -0.12 | 0.93 | 0.83 | 0.94 | 0.65 | -0.46 | -0.88 | -0.75 | -0.89 | -0.78 | 0.37 |
| | β-VAE | 0.94 | 0.97 | 0.96 | 0.53 | 0.54 | 0.86 | 0.83 | 0.89 | 0.78 | 0.79 | 0.17 | 0.13 | 0.23 | 0.87 | 0.90 |
| | DIP-VAE-I | 0.79 | 1.00 | 0.28 | 0.68 | 0.78 | 0.83 | 0.50 | 0.94 | 0.95 | 0.89 | 0.86 | 0.61 | 0.89 | 0.98 | 0.94 |
| | DIP-VAE-II | 0.93 | 0.92 | 0.81 | 0.42 | 0.46 | 0.53 | 0.13 | 0.76 | 0.99 | 1.00 | 0.49 | 0.03 | 0.72 | 0.99 | 0.99 |
| | FactorVAE | 0.87 | 0.92 | 0.80 | 0.23 | 0.11 | 0.65 | 0.14 | 0.68 | 0.79 | 0.89 | -0.38 | -0.79 | -0.30 | 0.66 | 0.64 |
| Shapes3D | AnnealedVAE | 0.95 | 0.62 | 0.21 | 0.91 | 0.96 | 0.60 | -0.13 | -0.51 | 0.91 | 0.83 | 0.87 | 0.45 | -0.01 | 0.97 | 1.00 |
| | β-TCVAE | 1.00 | 0.99 | 0.93 | 0.90 | 0.96 | -0.91 | -0.93 | -0.96 | -0.87 | -0.87 | -0.64 | -0.68 | -0.83 | -0.69 | -0.60 |
| | β-VAE | 0.99 | 0.97 | 0.86 | -0.57 | 0.92 | -0.53 | -0.62 | -0.80 | 0.94 | -0.55 | -0.57 | -0.66 | -0.84 | 0.92 | -0.63 |
| | DIP-VAE-I | 1.00 | 1.00 | 0.95 | -0.24 | -0.24 | 0.46 | 0.45 | 0.32 | 0.57 | 0.25 | -0.33 | -0.34 | -0.41 | 0.96 | 0.82 |
| | DIP-VAE-II | 0.99 | 0.99 | 0.97 | -0.74 | 0.74 | -0.63 | -0.78 | -0.83 | 0.98 | -0.44 | -0.73 | -0.86 | -0.90 | 0.97 | -0.56 |
| | FactorVAE | 0.35 | -0.02 | 0.03 | 0.49 | 0.44 | -0.80 | -0.96 | -0.90 | 0.59 | 0.43 | -0.85 | -0.98 | -0.96 | 0.45 | 0.27 |
| smallNORB | AnnealedVAE | -0.04 | 0.28 | 0.77 | 0.09 | 0.56 | -0.26 | -0.37 | 0.21 | 0.14 | 0.68 | 0.50 | 0.42 | 0.38 | -0.17 | 0.80 |
| | β-TCVAE | 0.97 | 0.76 | 0.98 | 0.94 | 0.99 | 0.86 | 0.53 | 0.97 | 0.78 | 0.92 | 0.93 | 0.64 | 1.00 | 0.89 | 0.99 |
| | β-VAE | 0.92 | 0.93 | 0.97 | 0.97 | 1.00 | 0.97 | 0.96 | 0.96 | 0.93 | 0.97 | 0.93 | 0.94 | 0.97 | 0.97 | 0.99 |
| | DIP-VAE-I | 1.00 | 0.98 | 0.95 | 0.97 | 0.98 | 0.97 | 0.94 | 0.91 | 0.93 | 0.94 | 0.99 | 0.98 | 0.96 | 0.97 | 0.98 |
| | DIP-VAE-II | 0.91 | -0.44 | 0.88 | 1.00 | 0.99 | 0.85 | -0.56 | 0.90 | 0.99 | 0.98 | 0.93 | -0.45 | 0.86 | 1.00 | 1.00 |
| | FactorVAE | 0.89 | 0.73 | -0.55 | 0.95 | 0.79 | -0.14 | -0.51 | 0.40 | 0.13 | 0.18 | 0.77 | 0.59 | -0.47 | 0.94 | 0.77 |

## F.4 Segmentation Task

The task involves segmenting the sprite from the Scream dSprites dataset using the learned representations, utilizing a decoder neural network that generates segmentation masks based on these representations. In Table 5, we list the results of these experiments. We observe that both $\overline{\text{IWO}}$ and $\overline{\text{IWR}}$ correlate stronger with downstream segmentation performance than classic disentanglement metrics.

## F.5 IWO Training

Given a learned representation of a dataset, we consider each generative factor independently, allocating separate LNNs respectively. On top of the LNNs, we have NN heads, which regress the generative factors from the intermediate projections. The NN heads are also independent from one another and do not share any weights.

### F.5.1 Implementation Details

We use the PyTorch Lightning framework[4] for the implementation of the models required to discern IWO and IWR. In particular, we use the implementations of the Linear and Batch-Normalization layers. Whereas the setup of the LNN is equal for all models and datasets, the NN-heads vary in their complexity for different datasets and factors. As

---

[4]https://github.com/Lightning-AI/lightning

Table 4: Correlation coefficients between IWO, IWR, DCI-D, DCI-C, MIG and the downstream task of recovering the generative factors with logistic regression and random forest. Examined latent representations of smallNORB dataset as learned by models: Annealed VAE (A-VAE), $\beta$-VAE, $\beta$-TCVAE and Factor-VAE (F-VAE) on 100%, 50% and 10% of the dataset

| | Model | $\overline{\text{IWO}}$ | | $\overline{\text{IWR}}$ | |
| | | Logistic Regression | Random Forest | Logistic Regression | Random Forest |
|---|---|---|---|---|---|
| 100% | A-VAE | 0.14 | 0.09 | 0.68 | 0.56 |
| | $\beta$-VAE | 0.93 | 0.97 | 0.97 | 1.00 |
| | $\beta$-TCVAE | 0.78 | 0.94 | 0.92 | 0.99 |
| | F-VAE | 0.13 | 0.95 | 0.18 | 0.79 |
| 50% | $\beta$-VAE | 0.20 | 0.02 | 0.43 | 0.75 |
| | A-VAE | 0.97 | 0.98 | 0.97 | 1.00 |
| | $\beta$-TCVAE | 0.82 | 0.95 | 0.94 | 0.99 |
| | F-VAE | 0.13 | 0.72 | -0.09 | 0.95 |
| 10% | A-VAE | -0.02 | -0.30 | 0.50 | 0.31 |
| | $\beta$-VAE | 0.94 | 0.99 | 0.96 | 0.99 |
| | $\beta$-TCVAE | 0.87 | 0.95 | 0.92 | 1.00 |
| | F-VAE | 0.32 | 0.56 | 0.12 | 0.58 |

Table 5: Correlation coefficients between IWO, IWR, DCI-D, DCI-C, MIG and the downstream task of segmenting the sprite in the Scream DSprite dataset.

| Model | DCI-D | DCI-C | MIG | $\overline{\text{IWO}}$ | $\overline{\text{IWR}}$ |
|---|---|---|---|---|---|
| Annealed VAE | 0.74 | 0.80 | 0.65 | 0.79 | 0.79 |
| $\beta$-TCVAE | -0.91 | -0.78 | -0.91 | -0.65 | 0.48 |
| $\beta$-VAE | -0.20 | -0.27 | -0.19 | 0.52 | 0.51 |
| DIP-VAE-I | 0.01 | 0.33 | -0.31 | 0.01 | 0.08 |
| DIP-VAE-II | 0.45 | 0.02 | 0.69 | 1.00 | 1.00 |
| Factor VAE | 0.09 | -0.33 | 0.19 | 0.63 | 0.62 |
| **Average** | 0.03 | -0.03 | 0.02 | 0.38 | 0.58 |

all considered models operate with a 10-dimensional latent space, each LNN has 10 layers. The output of each LNN layer is fed to the next layer and also to the corresponding NN-head.

Table 6 holds the NN head configuration per dataset and factor. These were found using a simple grid search on one randomly selected learned representation. This is necessary as factors vary in complexity and so does the required capacity to regress them. It is worth mentioning, that the Explicitness pipeline, as proposed by Eastwood et al. (2023), could actually be employed on top of the NN-heads, integrating both metrics.

For the initialization of the LNN layers and the NN heads, we use Kaiming uniform initialization as proposed in He et al. (2015). We further use the Adam optimization scheme as proposed by Kingma & Ba (2015) with a learning rate of $5 \times 10^{-4}$ and a batch size of 128 for all optimizations. Data is split into a training (80%) and a test set (20%). During training, part of the training set is used for validation, which is in turn used as an early stopping criterion. The importance scores used for IWO should be allocated using the test set. In our experiments, the difference between the importance scores computed on the training set and the test set was small. For further details on the implementation, please refer to our official code[5].

---

[5]https://github.com/cyrusgeyer/iwo

Table 6: Implementation Details for neural network heads operating on LNN layers. For each factor, ten NN heads with the specified number of hidden layers and their respective dimensions are trained in parallel.

| Dataset | Factor | Layer dimensions | Batch Norm | Factor Discrete |
|---|---|---|---|---|
| dSprites | Shape | 256, 256 | ✗ | ✓ |
| | Scale | 256, 256 | ✗ | ✗ |
| | Rotation | 512, 512, 512 | ✗ | ✗ |
| | x-position | 256, 256 | ✗ | ✗ |
| | y-position | 256, 256 | ✗ | ✗ |
| Cars3D | model | 256, 256, 256 | ✗ | ✓ |
| | rotation | 256, 256, 256 | ✗ | ✗ |
| | elevation | 256, 256, 256 | ✗ | ✗ |
| smallNORB | category | 256, 256 | ✓ | ✓ |
| | lightning condition | 256, 256 | ✓ | ✗ |
| | elevation | 256, 256 | ✓ | ✗ |
| | rotation | 256, 256 | ✓ | ✗ |
| Shapes3D | Floor color | 128, 128 | ✗ | ✓ |
| | Wall color | 128, 128 | ✗ | ✓ |
| | Object Color | 128, 128 | ✗ | ✓ |
| | Object Size | 128, 128 | ✗ | ✗ |
| | Object Type | 128, 128 | ✗ | ✓ |
| | Azimuth | 128, 128 | ✗ | ✗ |

Table 7: Average correlation between metrics and downstream task. Comparison between IWO, IWR and unweighted orthogonality (O)

| | Random Forest | | | Logistic Regression | | |
|---|---|---|---|---|---|---|
| Model | IWO | IWR | O | IWO | IWR | O |
| AnnealedVAE | 0.74 | **0.82** | 0.53 | 0.77 | **0.85** | 0.60 |
| $\beta$-TCVAE | **0.78** | 0.66 | 0.12 | **0.45** | 0.09 | 0.02 |
| $\beta$-VAE | 0.26 | **0.83** | -0.03 | 0.44 | **0.51** | 0.09 |
| DIP-VAE-I | 0.43 | 0.40 | 0.30 | **0.73** | 0.63 | 0.18 |
| DIP-VAE-II | 0.09 | **0.66** | -0.27 | **0.88** | 0.16 | 0.64 |
| FactorVAE | **0.68** | 0.61 | 0.59 | **0.71** | 0.69 | 0.56 |

## G  Importance Analysis

In order to test the effectiveness of IWO and IWR's importance weighing, we compare their downstream task correlation with the downstream task correlation of pure orthogonality (without any weighing). Table 7 holds the results of the correlation analysis. We can see, perhaps unsurprisingly, that the importance of weighing plays an important role in why IWO and IWR can serve as representation quality metrics, whereas orthogonality might be a questionable contender at best.

## H  Loss Analysis

In Figure 7, we depict the loss of neural network heads at different projection steps for a single run of synthetic experiment 3 ($L = 10$, $K = R = 5$). $\mathcal{L}_6$ to $\mathcal{L}_{10}$ are omitted, as they are almost zero (similar to $\mathcal{L}_5$). Each generative factor is analysed using GCA, which means for each generative factor we train an LNN spine with nine matrices $\boldsymbol{W}_9 \in \mathbb{R}^{9 \times 10}, \dots, \boldsymbol{W}_1 \in \mathbb{R}^{1 \times 2}$ and 10 NN heads acting on the projections. Because of the symmetry of the synthetic experiments, all five generative factors are similarly encoded in the latent space. In Figure 7, we are therefore depicting the mean and standard deviation over the generative factors. An entire pass through each LNN projects the representation to the most informative dimension for the respective generative factor. When recovering the generative factor from that projection, we incur a loss of $\mathcal{L}_1$. The fraction $\mathcal{L}_1/\mathcal{L}_0$ tells us that this is $\approx 20\%$ better than naive guessing (assuming the expectation value) of the factor. We see that we can almost perfectly recover the generative

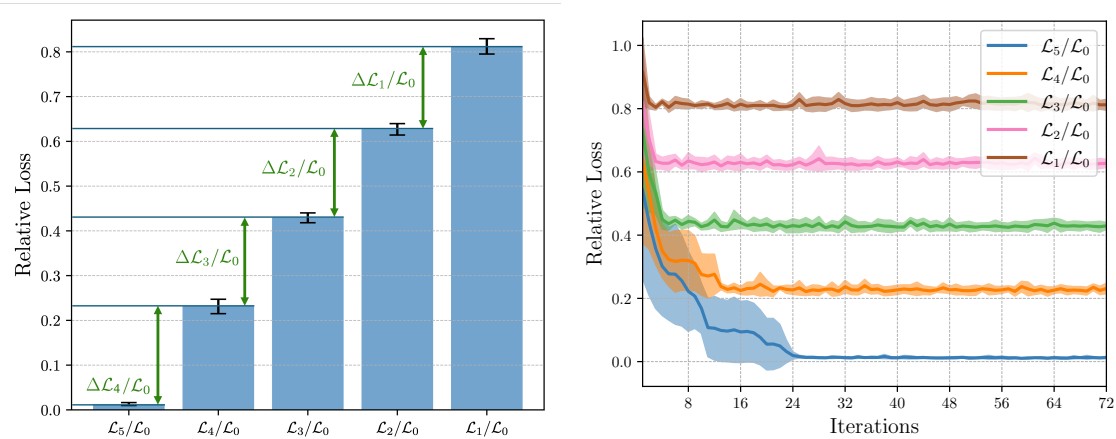

Figure 7: Loss of neural network heads at different projection steps for a single run of synthetic experiment 3 ($L = 10$, $K = 5$ and $R_j = 5$). We only depict $\mathcal{L}_5$ to $\mathcal{L}_1$, as $\mathcal{L}_6$ to $\mathcal{L}_{10}$ are almost zero (similar to $\mathcal{L}_5$). **Left:** Magnitude of relative loss after convergence for different projection depths. **Right:** Loss evolution over training iterations.

factors from projections to 5 and more dimensions. This is expected as the experiment is set up with $R_j = 5$. We see that each subsequently removed dimension increases the loss by $\approx 20\%$. It follows that $\Delta\mathcal{L}_l/\mathcal{L}_0 \approx 0.2$, i.e. $\alpha_l \approx 0.2$ for $1 \le l \le 5$. The right-hand side of Figure 7 depicts the relative validation loss of the NN heads during training. The relative loss at the 72nd iteration corresponds to the relative loss depicted in the left plot.

# I    Computational Resources Analysis

This section details the computational resources utilized for evaluating DCI, IWO and IWR, specifically applied to the smallNORB dataset from `disentanglement_lib` using a $\beta$-VAE framework. For the GCA model specifications please refer to section F.5.1

## I.1    Experimental Setup

- **Data:** Learned representations of a $\beta$-VAE trained on the smallNORB dataset from the *disentanglement_lib*
- **Objective 1:** Assess the orthogonality of 60 learned representations, by performing GCA and calculating IWO/IWR. Note that the computational resources for the calculation of IWO/IWR are negligible compared to GCA.
- **Objective 2:** Assess the disentanglement of 60 learned representations, by computing the DCI metric.

## I.2    Resource Utilization

In Table 8, we list the computational resources used for each run of GCA.

Table 8: Average resource utilization for each run of the GCA + IWO pipeline. No GPU was used.

| Resource | Usage |
|---|---|
| Process Memory | 400 MB |
| CPU Process Utilization | 50% |
| GPU Process Utilization | 0% |

## I.3    Runtime Analysis

- **GCA Average Duration:** 7 minutes per run (20 Epochs)
- **DCI Average Duration:** 4 minutes per run.

### I.4 Computational Cost Considerations for GCA

- For GCA computational costs scale with the number of linear layers in the LNN spine and the capacity of the NN heads.
- For large latent spaces, one should avoid step-wise dimensionality reduction in the LNN spine; larger reductions between consecutive LNN layers are preferred.
- The first LNN layer size need not match the dimensionality of the representation. For large representations, a smaller first LNN layer is recommended.
- GPU usage is beneficial for larger representations and models
- Performing GCA on pretrained smallNORB representations shows small computational costs, comparable to those necessary for computing DCI.

