# OpenReview forum: "Measuring Orthogonality in Representations of Generative Models"
_TMLR — Accepted by TMLR_

### Review · Reviewer_QLKU · 2024-07-22

**Summary Of Contributions:**

The paper tackles an extremely interesting topic: in the last decades, there has been an increasing interest in 'disentangled representations', with the hope to obtain 'better' representations, 'more interpretable'. A huge amount of empirical and theoretical work can now be found on disentangled representations. However, the practical interest in disentanglement remains to be proven. The paper brings empirical evidence that disentanglement might not be the right notion. The authors introduce two new metrics, IWO and IWR, based on the orthogonality of the latent factors, and show the proposed metrics correlate better with performance on downstream tasks.

**Audience:**

Yes

**Claims And Evidence:**

Yes

**Requested Changes:**

Clarify all the points of the previous section, in particular:
- Clarify the 'basis generation' part
- Detail the cost of the proposed method, and give insights on how much the hyperparameters need to be handtuned (as I understand, one has to learn multiple networks to compute the proposed metric)
- Comment more Table 1

**Strengths And Weaknesses:**

Overall it seems that the metric computation might be complex (and potentially costly) to compute, some parts of the paper need to be clarified. However, overall this is a very interesting paper, **challenging the correlation between disentanglement representation and performances on downstream tasks**.

Some parts of the paper were unclear to me:
- the General Component Analysis method has to learn a lot of neural networks, is the number of learned neural nets equal to the number of latent components?
- The 'subspace learning' part of 3.1 is very clear, but I did not understand the 'basis generation' part. Would it be possible to introduce a more rigorous/mathematical definition of the $Q_l$-s and  $q_l$-s. Is it clear that $Q_l$-s are full rank? i.e. that  $q_l$-s are uniquely defined? Is it even clear that the $Q_l$-s are of size $(l + 1) \time l$?
- What does $b_l$, $q_l$, $\Delta_l$ represent?
- where are the 'importance weights' $\alpha^{j}$ and $\alpha^{k}$ mathematically defined? Especially since the correlation applied between the latent can make the task especially easy (or hard)
- the synthetic data generation process could be explained more clearly, e.g, by writing the exact equation for each synthetic data.
- What are the conclusions of Table 1? I do not understand how it makes the point of the paper, and no explanation is provided. The only reference to Table 1 in the text is 'The results are displayed in Table 1'. Could you elaborate?
Is Table 1 displaying the correlation between downstream task performance and the metrics? If yes it seems to me that this is written nowhere.
- Results of Table 2 are extremely interesting! How do you understand the fact DCD-I, DCI-C and MIG correlate negatively with the performances for Random Forest?

---

> ### Author Response · Authors · 2024-09-11
> **Reply to Review**
>
> We thank the reviewer for their time and consideration.
>
> **Is the number of learned neural nets equal to the number of latent components?**
>
> Not necessarily. In the last paragraph of Section 4, we point out that higher reduction factors can be used when learning the matrices $W_l$. Additional details to speed up computation are presented in Appendix F.4 (we have now referred to them in the main paper).
>
> **Clarification about the basis generation subsection**
>
> Motivated by the reviewers remarks, we were able to further simplify the steps involved in the basis generation algorithm. We have modified Figure 2 according to these changes and added a Pseudocode implementation of the basis generation in Appendix E. We believe this simplification significantly increases the section’s clarity and thank the reviewer for inspiring these changes.
>
> **Clarification on notation**
>
> - $b_l^{(j)} \in {R}^L$: denotes a single basis vector of Basis $B_j$ spanning $z_j$’s latent subspace. In particular, $b_1$ represents the most important dimension capturing the variability of $z_j$, while $b_{R_j}$ is the least important one. We dropped the superscript $(j)$ in the basis generation section for ease of notation. Please refer to the enhanced section on basis generation for further details.
>
> - $dL_l^{(j)}$ is defined as the expected loss difference between two consecutive neural network heads: $d L_l^{(j)} = L_{l - 1}^{(j)} - L_l^{(j)}$ for $l > R_j$ (page 4), where $L_l^{(j)}$ is the expected loss of the neural network head $f_{jl}$ when regressing factor $z_j$ from the intermediate projection $w_l({c})$.
>
> - $a^{(j)}_l$ quantifies the importance of ${b}_l^{(j)}$ for regressing $z_j$ and is proportional to $dL_l^{(j)}$ as seen in equation 2 of our manuscript.
>
> - ${q}_l$: is removed, as we do no longer specify any specific type of basis completion in the basis generation section.
>
> We have included depictions of both $L^{(j)}_l$ and $\alpha^{(j)}_l$ into Figure 2, to better guide the reader through our methodology.
>
> **Clarification on synthetic experiments generation procedure**
>
> We have added pseudo-code for the synthetic data generation in Appendix E.2 and illustrated an example in Figure 5.
>
> **Conclusions of Table 1**
>
> The experimental results demonstrate that both $D$ and $C$ metrics from the DCI framework [1] exhibit a significant sensitivity to random orthogonal projections (ROP), rendering them inadequate for evaluating generative factor separability as we advocate it. While the $E$ metric from the DCI-ES [2] framework shows robustness to ROP, it nonetheless falls short in reliably capturing generative factor separability. This is because $E$ predominantly reflects the complexity of the mapping between generative factors, rather than their actual separability. In contrast, our proposed metrics, $\overline{\text{IWO}}$ and $\overline{\text{IWR}}$, accurately assess the orthogonality and importance distribution across the latent subspaces. They provide a more reliable evaluation of factor separability, demonstrating both resilience to orthogonal projections and immunity to mapping complexity. This reinforces the validity of our approach in contrast to existing methods.
> We have made sure to clarify these aspects in the respective section of our manuscript.
>
> **Negative Correlation of Disentanglement metrics with Random Forest Performance.**
>
> We attribute the negative correlation between disentanglement metrics and downstream random forest performance on the Cars3D dataset to the inherent complexity of the generative factors within the dataset. For instance, the Cars3D dataset comprises 184 distinct car models, each introducing considerable complexity. To achieve high disentanglement, a model would need to represent all these variations in a subspace aligned with the canonical basis, often requiring linear relationships.
>
> Consider two identical linear representations, only one is axis-aligned, while the other is linearly projected such that its axis alignment is disrupted. In this case, a random forest will naturally favor the former due to the simplicity of axis alignment. However, when comparing an axis-aligned linear representation to a non-axis-aligned, non-linear representation, which, however, encodes a significantly higher amount of predictive information, the random forest might perform better on the latter. With sufficient trees, the random forest can learn complex patterns even if the representation is non-axis-aligned, non-linear.
>
> In summary, while a random forest generally favors axis-aligned representations when other characteristics are kept equal, this preference may reverse when the constraints of linearity and axis alignment severely limit the predictive potential of the representation.

---

> > ### Author Response · Authors · 2024-09-11
> > **Reply to Review [2/2]**
> >
> > **Cost of the proposed method**
> >
> > We would like to direct the reviewer to Section F.5.1 Implementation Details and Section I Computational Resource Analysis of our Appendix. In Section F.5.1, we provide an in-depth explanation of the hyperparameter search and additional experimental details. Section I offers a detailed breakdown of the computational costs. In summary, the computational resources required for assessing $\overline{\text{IWO}}$ and $\overline{\text{IWR}}$ are rather small, underscoring the efficiency of our proposed metrics.
> >
> > We thank the reviewer very much for their constructive feedback and remain open to further discussions,
> >
> > Kind regards,
> >
> > The Authors
> >
> > [1] Eastwood, Cian, and Christopher KI Williams. "A framework for the quantitative evaluation of disentangled representations." 6th International Conference on Learning Representations. 2018.
> >
> > [2] Eastwood, Cian, et al. "Dci-es: An extended disentanglement framework with connections to identifiability." arXiv preprint arXiv:2210.00364 (2022).

---

### Review · Reviewer_UrTf · 2024-08-07

**Summary Of Contributions:**

This paper proposed two metrics, Importance-Weighted Orthogonality (IWO) and Importance-Weighted Rank (IWR), for measuring the orthogonality of representations such that each factor is not required to be aligned with the basis. The paper argues that these metrics correlate better with downstream task performance than traditional disentanglement metrics. The proposed metrics were evaluated on synthetic datasets.

**Audience:**

Yes

**Claims And Evidence:**

Yes

**Requested Changes:**

- Locatello et al. (2019) used not only orthogonal transformations but also probability integral transforms and their inverses.
- p. 2 "we utilize Generative ..." sounds like "Generative Component Analysis (GCA)" was proposed in a previous work.
- As far as I know, the original paper on the group-theoretical definition (Higgins et al. 2018) never used the term "Symmetry-Based Disentanglement Representation".

**Strengths And Weaknesses:**

## Strengths

- The motivation of this paper is clear, and the author proposed a reasonable solution.
- This paper is contextualized. The related work section is useful for the readers.
- The author proposed "Generative Component Analysis (GCA)" to identify the subspaces where generative factors vary.

## Weaknesses

- Given a metric, I think it is important to state clearly:

  - what property it exactly measures,
  - **if having a perfect score (either $0$ or $1$) implies that a representation must have the property, and if all representations with the property are assigned perfect scores**,
  - what supervision is needed to calculate the metric,
  - if it is differentiable so that we can directly optimize the property,
  - what supervision/annotation is needed, and
  - the computation cost of the metric.

  The author mentioned some of them, but it would be better to summarize them as theorems (with proofs).

- The author explained the design choice of the proposed metrics, but not the reasons. For example, the benefit of "taking the importance of the dimensions spanning them into consideration" is unclear to me.

---

> ### Author Response · Authors · 2024-09-11
> **Reply to Review [1/2]**
>
> We thank the reviewer for their time and consideration.
>
> # Addressing Weaknesses
>
> **What properties do the metrics measure**
>
> Given two generative factors $z_j$ and $z_k$:
> - $\text{IWO}$ measures the pairwise weighted projection between the latent subspaces where the factors vary. The weighing is determined by the predictive power (importance) of the dimensions spanning the respective latent subspaces.
> - $\text{IWR}$ measures how much the predictive power (importance) of a generative factor is spread across the representation space.
>
>
> $\overline{\text{IWO}}$ and $\overline{\text{IWR}}$ refer to the average of IWO and IWR over all generative factors of a dataset.
>
> Both metrics rely on GCA, which allocates these latent subspaces by finding importance ordered orthogonal basis spanning them. Each basis vector is allocated an importance weight proportional to how much predictive power w.r.t the generative factor the respective dimension encompasses.
>
> **Lowest and highest scores**
>
> We have added proofs in the appendix of our manuscript showcasing that, given a representation $\mathbf{c} \in \mathbb{R}^L$ with $L > 1$:
> - $\overline{\text{IWO}} = 1$ if and only if the latent subspaces $\mathbb{S}_1, \dots, \mathbb{S}_K$ of the generative factors $z_1, \dots, z_K$ all lie in each other's orthogonal complement.
> - $\overline{\text{IWO}} = 0$ if and only if the latent subspaces $\mathbb{S}_1, \dots, \mathbb{S}_K$ of the generative factors $z_1, \dots, z_K$ all share the same importance along the same dimensions.
> - $\overline{\text{IWR}} = 1$ if and only if the latent subspaces $\mathbb{S}_1, \dots, \mathbb{S}_K$ of the generative factors $z_1, \dots, z_K$ are all uni-dimensional.
> - $\overline{\text{IWR}} = 0$ if and only if for each generative factor $z_1, \dots, z_K$ the importance is spread equally among all dimensions of the representation space.
>
> **Supervision**
>
> $\text{IWO}$ and $\text{IWR}$ require access to the true data generating factors. This type of supervision has been the de-facto standard in evaluating disentanglement, and most disentanglement metrics, from $\beta-VAE$ [1] up to recent contributions such as $DCI-ES$ [2], rely on it. We have made sure that this is stated clearly in the manuscript.
>
> **Differentiability**
>
> GCA and both$\text{IWO}$ and $\text{IWR}$ are composed of differentiable operations only, which
> makes the metrics differentiable. We have added this information in Appendix F of the manuscript.
>
> **Direct optimisation and supervision**
>
> While this paper is concerned with showing that Orthogonality is better suited for measuring representation utility in *unsupervised* learning compared to Disentanglement, following the lines of Fair VAE [3] and Flexible Fair VAE [4], it would certainly be possible to directly optimise the properties measured by IWO and IWR through access to the ground truth generative factors during *supervised* training in follow up work.
>
> **Computation Cost**
>
> For GCA, the subspace learning step is dominated by the training of the neural network heads, therefore by the backpropagation algorithm used, while the basis generation step is dominated by the basis completion algorithm used (e.g. QR-decomposition).
> As noted in the experimental section, we can reduce the number of  neural network heads by compressing multiple dimensions, and additional computational efficiency and speed-up details can be found in Appendix I (cf. also answer to reviewer QLKU). After performing GCA, the worst case complexity of $\overline{\text{IWO}}$ is $O(KL^3)$, with $K$ being the number of generative factors. For $\overline{\text{IWR}}$, the worst case complexity is $O(KL)$. We have included this discussion in the paper.
>
> **Motivation of design choices**
>
> We have made sure that we motivate our design choices throughout our manuscript more rigorously, in particular also the section pointed out by the reviewer. We thank the reviewer for pointing this out.
>
> # Requested Changes
>
> We thank the reviewer for pointing us to these parts of our manuscript. We have added the use of integral transforms by Locatello et al. and have rephrased the sentence about GCA to avoid any misinterpretation. Symmetry-Based Disentangled Representation Learning (SBDRL) is a term borrowed from Caselles-Dupré et al. (2019) [5], since Higgins et al. did not provide a name for their methodology. We have however replaced it with a more appropriate terminology.
> We thank the reviewer again for their time and consideration and hope that our rebuttal and the changes made to our manuscript address their remarks satisfactorily.
>
> We remain open for further discussion.
>
> Thank you and best regards,
>
> The authors.

---

> > ### Author Response · Authors · 2024-09-11
> > **Reply to Review [2/2] – References**
> >
> > [1] Burgess, Christopher P., et al. "Understanding disentangling in $\beta $-VAE." arXiv preprint arXiv:1804.03599 (2018).
> >
> > [2] Eastwood, Cian, et al. "Dci-es: An extended disentanglement framework with connections to identifiability." arXiv preprint arXiv:2210.00364 (2022).
> >
> > [3] Louizos, Christos, et al. "The variational fair autoencoder." arXiv preprint arXiv:1511.00830 (2015).
> >
> > [4] Creager, Elliot, et al. "Flexibly fair representation learning by disentanglement." International conference on machine learning. PMLR, 2019.
> >
> > [5] Caselles-Dupré, Hugo, Michael Garcia Ortiz, and David Filliat. "Symmetry-based disentangled representation learning requires interaction with environments." Advances in Neural Information Processing Systems 32 (2019).

---

### Review · Reviewer_YKwd · 2024-08-20

**Summary Of Contributions:**

The paper proposes two new metrics, Importance-Weighted Orthogonality (IWO) and Importance-Weighted Rank (IWR), to evaluate the disentanglement of representations, as the existing methods fail to address the issue in Fig. 1, where some certain bases are better than others to capture the disentanglement. Through extensive experiments, the paper shows that these metrics correlate more strongly with downstream task performance compared to traditional disentanglement metrics. The paper's finding indicates that the utility of a representation is closer related to the orthogonality than its disentanglement.

**Audience:**

Yes

**Broader Impact Concerns:**

Not applicable.

**Claims And Evidence:**

Yes

**Requested Changes:**

Explain which metric should be used to measure the representation quality when a downstream task is not known or cannot be trivially evaluated.

More tasks should be tested to demonstrate the representation quality.

The diverse results in Table 2 should be discussed more.

**Strengths And Weaknesses:**

Strengths:

The perspective that disentanglement is a better measurement for feature representation is interesting.

The proposed Generative Component Analysis (GCA) and IWO/IWR are empirically validated to be effective in most experimental settings.


Weaknesses:

In Table 2, it shows that IWO and IWR have distinct performances in some cases, which raises the question: which metric should be used to measure the representation quality when a downstream task is not known or cannot be trivially evaluated?

Only simple classification tasks are tested to show the representation quality. It is unknown how the representation is useful in different tasks like object detection, segmentation, etc.

In Color DSprites, all disentanglement metrics fail to correlate with the representation utility, and there is no discussion on this. Similarly, in Randon Forest, IWR/IWO is not as good as DCI-C in Table 2b.

---

> ### Author Response · Authors · 2024-09-11
> **Reply to Review**
>
> We thank the reviewer for their time and consideration.
>
> **Choosing between IWO and IWR**
>
> The choice between using IWO or IWR depends on several factors, particularly the complexity of the dataset and the downstream model. This relationship is analogous to how Disentanglement relates to Completeness in the DCI framework proposed by Eastwood et al.
> Our experiments indicate that the correlation between these metrics and downstream task performance varies depending on the complexity of the dataset and the model used. Specifically:
> - For simpler models and datasets, IWR tends to perform better or as well as IWO. For instance, when using a Random Forest, IWR appears to assess representation utility effectively across most datasets. The exception is the cars3D dataset, which features 184 different car models and thus presents a higher level of complexity.
> - For more complex models or datasets, IWO generally becomes the better choice. This suggests that IWO may be more robust in scenarios where the task or data complexity increases.
> In summary, if simplicity is a priority—for example, to ensure interpretability or due to resource constraints—IWR can be a valuable metric. However, in most other cases, particularly when dealing with higher complexity, IWO is the more appropriate choice.
>
> **Other Downstream Tasks**
>
> We acknowledge the value of evaluating representation quality on a broader range of tasks, such as object detection and segmentation. While we have adhered to the standard benchmarks commonly used in disentanglement research to ensure reproducibility and comparability with existing work, we agree that expanding the scope to include additional tasks can provide further insights.
>
> To address the reviewer’s suggestion about additional, more complex tasks, we have designed new experiments that incorporate a segmentation task into our evaluation. We have chosen the Scream dSprites datasets for this purpose, as this dataset implicitly includes segmentation masks through the raw binary dSprite data, on which basis it is generated. Refer to Figure 6 and section F.2 in the Appendix for additional details. The task involves segmenting the sprite inside the images using only the learned representations, by means of a decoder neural network that generates segmentation masks. In the following, we list the results of these experiments, which we have also included in Appendix F.4.
>
> | Model| DCI-D |DCI-C|MIG|IWO|IWR|
> |--------------|----------|----------|----------|----------|----------|
> |Annealed VAE |0.74|0.80|0.65|0.79|0.79|
> |$\beta$-TCVAE|-0.91|-0.78|-0.91|-0.65|0.48|
> |$\beta$-VAE|-0.20|-0.27|-0.19|0.52|0.51|
> |DIP-VAE-I| 0.01| 0.33|-0.31|0.01|0.08|
> |DIP-VAE-II|0.45|0.02| 0.69| 1.00| 1.00|
> |Factor VAE| 0.09| -0.33| 0.19|0.63|0.62|
> |**Average**|**0.03**|**-0.03**|**0.02**| **0.38**| **0.58**|
>
> We can see that both IWO and IWR correlate stronger with downstream segmentation performance than classic disentanglement metrics.
>
> We also considered object detection, however many of the datasets we employ already encompass aspects of this task. Specifically, factors such as the position, size, rotation, and type of objects depicted are integral to the generative factors within these datasets (see Appendix F.2 for details).
>
>
> **Discussion about correlation**
>
> First, regarding the Color dSprites dataset, we observe a positive correlation between disentanglement metrics and downstream task performance across Random Forest and Logistic Regression models. It is only for the Multi-Layer Perceptron that we see a negative correlation. This suggests that the behaviour observed is task-dependent, and not a general characteristic of the dataset itself. We will be happy to further expand this analysis in the manuscript to underline this important distinction.
>
> Second, regarding the performance of IWR/IWO in comparison to DCI-C in Table 2b, we acknowledge the reviewer’s observation that DCI-C correlates better with Random Forest performance. This can be attributed to the fact that DCI-C directly measures axis alignment, which is highly relevant to tree-based models like Random Forests. As we note in the manuscript, “the downstream task benefits from the alignment of the generative factors with the canonical basis.” This alignment directly impacts the performance of models that rely on axis-aligned splits, such as random forests. In contrast, IWO/IWR, which emphasize orthogonality, do not necessarily prioritize this form of alignment, which explains their weaker correlation with Random Forest performance.
>
> In conclusion, while we do not claim that a single, universally optimal quality metric exists for representation learning, we argue that orthogonality offers a more robust measure of representation quality for a broader range of downstream tasks, as in many scenarios disentanglement alone does not capture the full picture of a representation’s utility.
>
> We remain open for further discussion.
>
> Kind regards,
>
> The authors

---

### Decision · Action_Editor_QUso · 2024-10-01

**Recommendation:** Accept as is

**Comment:**

Post-discussion, all three reviewers are in favor of acceptance and are satisfied with the revised version of the paper. The AE is in favor and recommends accepting the paper.

**Audience:**

Yes. The topic is of clear interest.

**Claims And Evidence:**

Yes. All three reviewers agree and the AE concurs.